# Political development predicts reduced human cost of flooding

**Paola Vesco** [1,5] ✉, **Nina von Uexkull** [1,2,3,5], **Jonas Vestby** [1] & **Halvard Buhaug** [1,4,5]

Societal impacts from extreme climate and weather events depend not only on hazard magnitude but also on the vulnerability of the affected population. Existing research suggests that adverse socioeconomic conditions are associated with higher baseline vulnerability to many types of risk, but comparatively little attention has been paid to political drivers of vulnerability. Focusing on floods, the most frequent climate-related hazard, this article evaluates the impact of political development on flood mortality. Findings from a Bayesian predictive analysis of global flood impacts from 2000 to 2018 suggest that democracy, institutional quality, and peace reduce the predicted human cost of flooding. The effect of a breakdown of peace on predicted flood mortality is especially pronounced. These results indicate that promoting peace, justice, and strong institutions (Sustainable Development Goal 16) can help to mitigate disaster risks and support effective climate change adaptation.

Extreme weather-related events are on the rise. Since 1980, climatological, meteorological, and hydrological hazards have resulted in more than 11,000 disasters that collectively have claimed 1.6 million human lives worldwide[1] (Fig.1a). The growth in recorded weather-related disasters has been attributed to anthropogenic warming[2,3], although it also is a result of population growth in hazard-prone areas and improved quality of reporting over time[4,5].

Disaster impact is a function of hazard severity, magnitude of exposure, and level of vulnerability[6,7]. Despite the steep growth in human exposure to weather-related hazards, global disaster mortality has been on a long-term decline, epitomizing considerable success in reducing social vulnerability to climate extremes (Fig. 1b). However, progress has been uneven within and across world regions. Low economic growth accentuated by the COVID-19 pandemic, escalating external debt rates, an increasingly polarized geopolitical context, and unabated global warming constitute fundamental challenges to further progress. Evidence of stagnation and even reversal of recent gains in disaster risk reduction is mounting[8,9].

Against the backdrop of these challenges, there is urgent need for effective vulnerability-reducing adaptive responses[10]. However, scientific knowledge of what generates vulnerability remains inadequate. Most empirical research is location-specific and concerned with individual- and community-level factors, e.g. refs. 11–20. There is comparatively limited systematic evidence of macro-level conditions that shape disaster risk[21,22]. Likewise, contemporary assessments and policy discourses privilege a focus on technical, financial, legal, and behavioral solutions over addressing barriers related to the broader political context[9,23–26]. Yet, the potential for and viability of specific risk management interventions is also a function of the interests and capacities of national governments that hold primary responsibility for disaster risk reduction[27,28]. Vulnerability to climate hazards is not unavoidable, but is co-produced by existing political power structures[29–31]. Knowledge of specific political drivers of vulnerability thus is directly relevant for informing priorities in disaster risk reduction.

To address this research gap, we investigate how political development moderates the human cost of disaster at global scale, 2000–2018, drawing on a theoretically-driven predictive analysis. We evaluate the contribution of political development indicators to predicting flood mortality both in-sample and on unseen data, with

[1]Peace Research Institute Oslo, Oslo, Norway. [2]Department of Peace and Conflict Research, Uppsala University, Uppsala, Sweden. [3]Department of Politics and Public Administration, University of Konstanz, Konstanz, Germany. [4]Department of Sociology and Political Science, Norwegian University of Science and Technology, Trondheim, Norway. [5]These authors contributed equally: Paola Vesco, Nina von Uexkull, Halvard Buhaug. ✉e-mail: paoves@prio.org

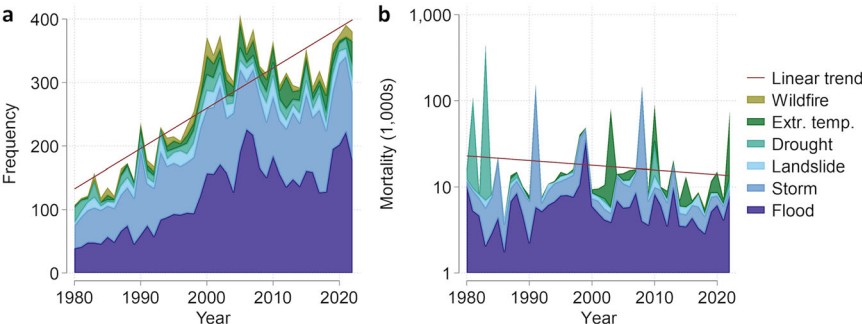

**Fig. 1 | Global trends in weather-related disasters, 1980–2022. a** Annual frequency of disasters by type ($N = 11,428$). **b** Annual mortality of disasters by type ($\log_{10}$). Red lines denote linear global trends. Data source: EM-DAT[1]. A disaster fulfills one or more of the following criteria: At least 10 people killed; at least 100 people affected; declaration of a state of emergency; or call for international assistance.

alternative empirical strategies adopted in robustness tests. We focus on three key dimensions of political development: (i) democratic characteristics of formal political institutions ('democracy'), (ii) the capacity and performance of these institutions ('institutional quality'), and (iii) peaceful modes of conflict resolution ('conflict')[32,33]. This framing is consistent with common definitions of 'good governance'[34,35] and it also directly relates to Goal 16 of the 2030 Agenda for Sustainable Development, which targets effective, accountable, and inclusive political institutions and the promotion of peace (SDG 16)[36].

The three dimensions of political development are theorized to influence disaster risk reduction (DRR) through a number of channels. First, accountable and inclusive power-sharing institutions, which are essential features of democracy, facilitate impartial politics, which is important to secure the rights, representation, and protection of vulnerable social groups[37,38]. Transparent decision-making, regulated competition, and a free and independent press further serve as checks on the exercise of political power, including in disaster risk reduction[28,39,40]. All else equal, democratic governance structures create greater political incentives to leave no one behind in disaster preparedness and response. Conversely, vulnerability to disaster impacts is often especially high where people are excluded from decision-making processes[41–43].

Second, high quality of political institutions, which is associated with high bureaucratic competence and effectiveness, entails greater ability to mobilize resources for public goods, more efficient government spending, and superior capacity to coordinate between governmental agencies and civil society actors when crises occur[44,45]. High bureaucratic capacity also is associated with deeper societal penetration of the central state, consistent enforcement of rules and regulations, and control on corruption, which jointly result in higher levels of political trust and public compliance[46,47]. In societies where the state is fragile or de facto absent, disaster impacts often spike[38,48].

Third, a "Weberian" state[49], characterized by functioning political and legal institutions for peaceful conflict management, fosters social cohesion and stability required for long-term planning and economic investments as well as effective development assistance[50,51]. A breakdown of societal peace, in contrast, erodes social trust, polarizes or paralyzes civil society, biases political attention and prioritization, damages or destroys critical infrastructure, and generates widespread human suffering[52]. For people exposed to armed conflict, vulnerability to external shocks increases through, e.g., loss of income and assets, compromised health and food security, broken social ties, and forced displacement[53–56]. Ongoing conflict further complicates or obstructs development aid and disaster relief operations[57,58]. More generally, armed conflict represents a major barrier to sustainable development[8].

Although it is instructive to separate between dimensions of political development conceptually, they often go together empirically. For example, peace is a necessary precondition for effective institutions and inclusive governance[59] whereas democracy fosters nonviolent modes of conflict resolution[60]. Political development also influences – and is influenced by – economic performance, whereby democracy and high institutional quality generally are associated with greater economic stability and growth, shaping countries' development trajectories[61–64]. Closing the endogeneity circle, political institutions and peace can be sensitive to the impacts of severe climate hazards, especially in vulnerable contexts[65–67].

## Results
We assess the role of political development in affecting human cost of disaster via in-sample regression analysis as well as in-sample and out-of-sample prediction validation. The endogenous relationship and typically slowly-changing nature of political development present considerable challenges to inference from conventional regression analysis, which lead us to use predictive modeling as our main empirical strategy, with other approaches for robustness documented in Supplementary Information. The analysis focuses on the most frequent disaster type in modern times: floods[68]. Although a society's vulnerability can vary considerably across hazard types, key country-level features, such as the characteristics and capacity of political institutions, shape the underlying propensity to harm from many forms of risks, including war, pandemics, and natural hazards[10,69,70]. Thus, while this study does not directly speak to human impacts from other disaster types, results are likely to be broadly representative of macro-level political drivers of climate-related vulnerability.

Specifically, we exploit high-resolution data on the timing and spatial extent of major floods worldwide, 2000–18, provided by the Global Flood Database (GFD)[71]. We exclude floods that occur in uninhabited areas and treat floods that span across international borders as separate events in each country, resulting in a valid sample of N = 2,225 unique flood-country events. The outcome variable is the number of reported fatalities for each event, obtained from the accompanying Dartmouth Flood Observatory (DFO) archives[72]. By structuring our analysis around individual flood events, as opposed to comparing flood and non-flood observations in a panel data structure, we ask: given flooding, to what extent do political factors mitigate loss of human lives?

We rely on established data sources to operationalize the three dimensions of political development, combining country-level statistics with georeferenced data to capture the local context in the flooded areas. Democracy is represented by indices of governmental accountability[35] and degree of inclusion of social groups in society[73]. Institutional quality likewise is represented by two indices: governmental effectiveness[35] and rule of law[73]. We measure the third political dimension by focusing on the breakdown of peace: a country-level indicator of recent civil conflict-related battle-deaths[74], supplemented

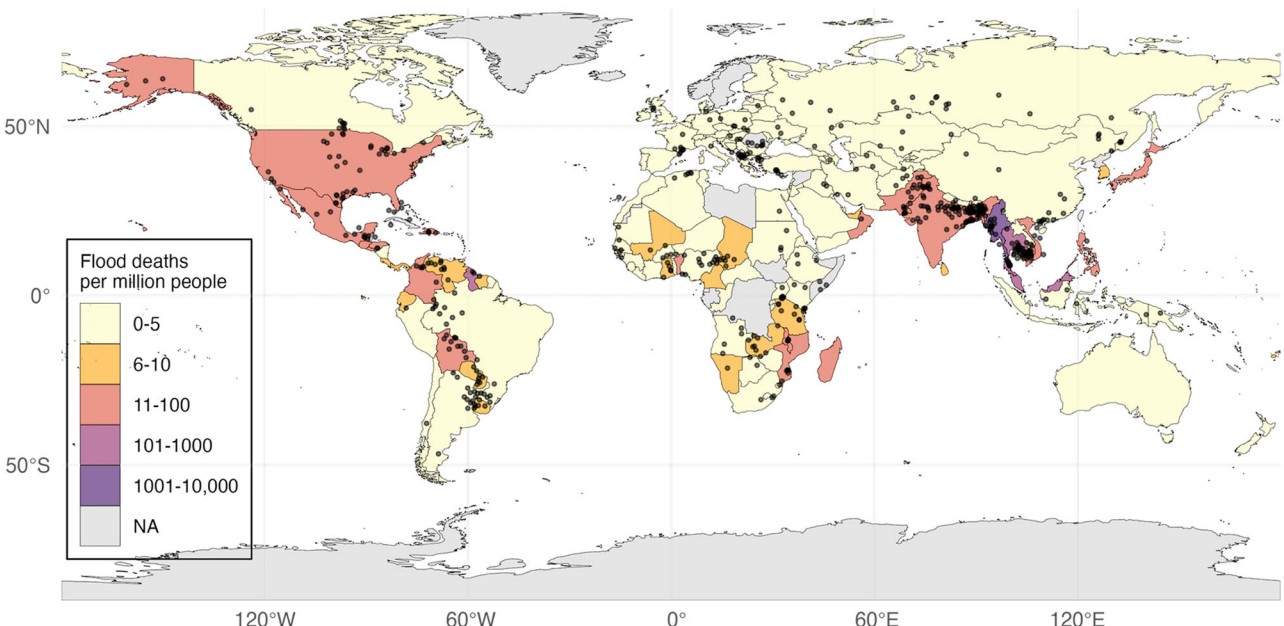

**Fig. 2 | Map of flood location and severity by country, 2000–18.** The dots denote the centroids of the flooded areas of all GFD events included in the analysis ($N = 2225$). Country colors reflect a simple categorization of total flood deaths per million people in the period, from yellow (low) to purple (high impact). Gray territories have missing information (i.e., there are no recorded flood events in the Global Flood Database or the flood rasters do not overlap with the georeferenced human settlement data).

by a measure of local conflict exposure within the flooded areas during the previous 12 months[75]. Since these indicators correlate (Supplementary Fig. S2-S3), we assess their influence on the outcome in sequential models.

In addition, all models contain a set of baseline predictors of flood mortality. Characteristics of each flood event include hydrological severity of the flooding, duration of the flood in days, a binary indicator for tropical storm-related floods, and a count of flood events in the country over the previous decade, all generated on the basis of DFO[72]. Estimates of population exposed to flooding were obtained from[76]. Lastly, the models in the main specification include an index of terrain ruggedness[77], as well as indicators of socioeconomic development, to increase coherence with the existing literature[78]. As socioeconomic factors may be 'bad controls', however, we document results from models that exclude socioeconomic development in Supplementary Information Section 2.2.

The global prevalence and country-level human impact of flood events recorded by the GFD are visualized in Fig. 2. While no region is exempt from flood risk, severe impacts concentrate in the tropics, notably Southeast Asia.

## Indicator performance

We begin the empirical evaluation by assessing in-sample conditional effect plots for the political indicators, derived from Bayesian negative binomial regression with 32,000 samples across 8 chains for each model. Results are largely in line with expectations. Flooding in countries with accountable (Fig. 3a) and high-quality (Fig. 3b, e) political institutions is estimated to claim fewer lives than floods in countries with weaker institutional characteristics, all else equal. The influence of inclusion (Fig. 3d) is comparatively weaker but in the expected direction, where predicted flood impact is lower in more inclusive societies. Breakdown of peace has a powerful influence on the outcome; severe conflicts at the country level as well as in the flooded areas are associated with substantially higher flood mortality estimates (Fig. 3c, f). This effect is likely to be conservative: as conflicts hamper socioeconomic development[79], the predicted effect of conflict is at least partly absorbed by GDP per capita and local HDI (see also

Supplementary Fig. S9). Although the predicted effect of democratic characteristics and quality of institutions on median flood mortality (black line in Fig. 3) is comparatively weaker than the predicted effect of conflict, the 80% predictive intervals (shaded areas, Fig. 3) indicate that all six political development indicators explored here have substantively important impacts at the extremes of the flood mortality distribution. This is consistent with our expectation that more democratic, efficient, and peaceful countries are considerably better than other political regime types at preventing flooding from escalating into humanitarian disasters.

Estimated conditional effects for baseline predictors (Fig. 3g–n) are largely as expected. All else equal, predicted flood mortality increases with hydrological flood severity, population exposure and frequency of recent flood events, and mortality also is positively associated with short-duration (flash) floods. Local HDI decreases predicted flood deaths, while National GDP per capita is associated with higher predicted flood mortality. This somehow counter-intuitive result is likely driven by the high correlation (>0.8) between the socioeconomic indicators in our dataset (Supplementary Fig. S2); when local HDI is omitted from the baseline, the effect of GDP is as expected (Supplementary Fig. S14). The effects of terrain ruggedness and tropical floods on flood mortality are comparatively weaker.

## Model performance

Structural socioeconomic and political features are connected to each other in complex, endogenous ways, and they also shape and are informed by other societal features, challenging inference from conventional regression analysis. We therefore assess the informational value of political factors in predicting flood mortality, arguing that their predictive power, alongside theoretical reasoning, offers the strongest basis for understanding flood mortality drivers given the available data[80]. Specifically, we evaluate overall model performance both in-sample and out-of-sample, relying on two performance metrics: expected log predictive density (elpd), which measures the fit of a model's predictive distribution relative to observed outcomes[81], and stacking, which assesses the relative performance of each model compared to all other models considered[82]. We estimate elpd and

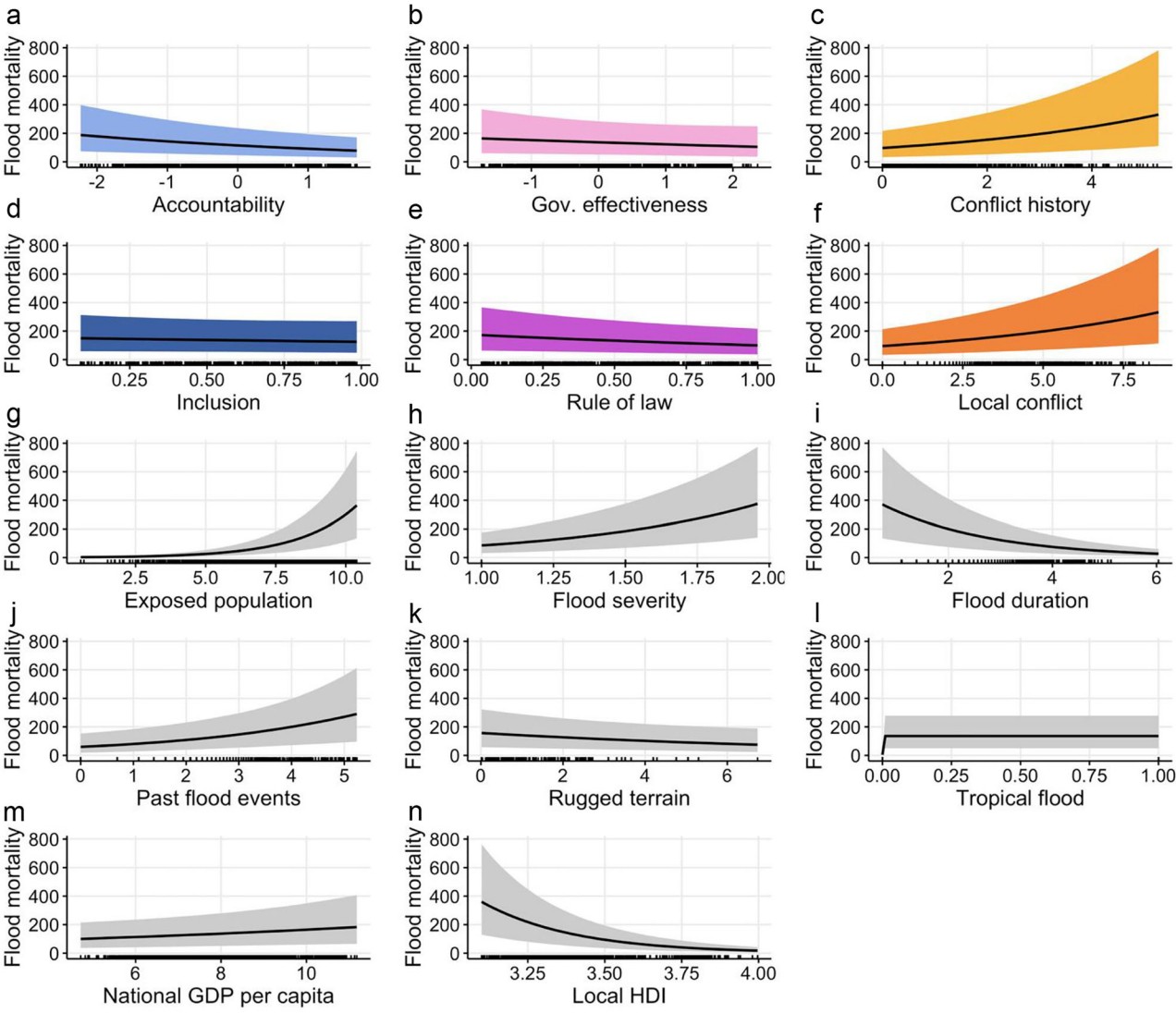

**Fig. 3 | Conditional effect plots for political drivers of flood mortality.** Each plot (**a**–**n**) shows the in-sample median of the posterior predictive distribution (black line) and the surrounding 80% predictive interval (shaded area) for the selected indicator, based on 8 × 4000 Markov chain Monte Carlo draws per model. Rug plots display the distribution of data points. All effects are computed by including the selected indicator of political development only, in addition to the baseline predictors (gray), holding other variables at their observed mean values. Colors reflect political development dimensions: blue shades for democracy indicators (accountability, inclusion), pink for institutional quality indicators (gov. effectiveness, rule of law), orange for peace breakdown indicators (conflict history, local conflict). To aid visual interpretation, the plots are shown for a subset of observations excluding the 20% most severe conflict history events, although the underlying models were estimated on the complete training sample, 2000–14 ($N = 1914$).

stacking weights both in-sample (i.e., the predictions are evaluated against the observations used to train the model) and out-of-sample, mimicking early warning analysis. Training and test sets differ in terms of countries included (Supplementary Fig. S4) and related sample characteristics (Supplementary Tables S1, S2). Notably, average mortality per flood event in the test set is only 1/10th of the training sample. Out-of-sample validation therefore is a particularly hard test in this case. Figure 4a shows the normalized in-sample and out-of-sample elpd scores for the alternative political development models and the baseline model.

As anticipated, the inclusion of conflict significantly enhances predictive performance; the two conflict models outperform all other models in-sample, and have the highest predictive performance out-of-sample along with government effectiveness. The relative importance of conflict is also reflected in the stacking weights, where the large majority of observations are best predicted by the conflict models. The accountability model gets 40% of the stacking weight in-

sample, indicating its superior predictive capability for a significant share of the flood observations in the training set (Fig. 4b). Together, these results testify to the substantive importance of accountable and effective institutions, beyond peaceful modes of conflict resolution. However, we note that the difference in elpd values, especially between the non-conflict models, is modest and characterized by high uncertainty, reflecting that the indicators are correlated and partly substitute for each other (Fig. 4c).

To assess the robustness of these results, we conducted a series of sensitivity tests. In brief, we implemented the following changes: (i) replaced the political predictors with aggregate political development indices; (ii) dropped socioeconomic controls from the baseline model; (iii) introduced interactions between the political development indicators and GDP per capita; (iv) excluded subnational predictors; (v) dropped continental random effects and time trends; (vi) limited training and test samples to floods with at least 1000 people exposed; (vii) limited samples to fatal floods; (viii) employed alternative training

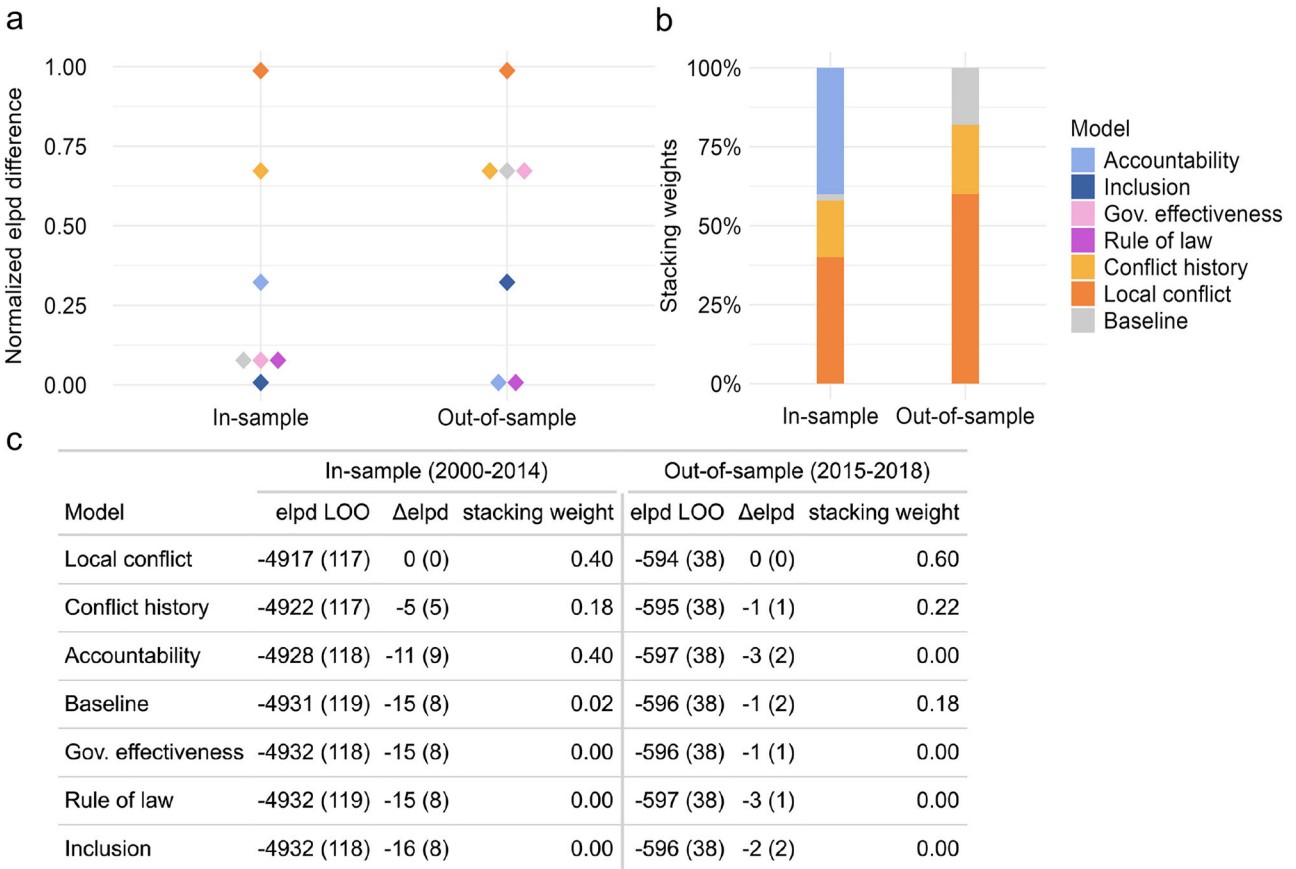

**Fig. 4 | Relative predictive performance of political development models on flood mortality. a** In-sample (2000–14, $N = 1914$) and out-of-sample (2015–18, $N = 311$) expected log predictive density (elpd) for each indicator model obtained via leave-one-out cross-validation with Pareto-smoothed importance sampling, normalized such that the most influential model scores 1 and the least influential model scores 0. **b** In-sample and out-of-sample stacking weights for each indicator model, reflecting the share of observations for which a given model's predictive performance is superior to all other models. **c** In-sample and out-of-sample elpd and stacking weights for each indicator model. Models are ranked by in-sample elpd performance (values closer to zero are better; standard errors in parentheses). Each color denotes a model based on the political development dimension captured by the main predictor: blue for democracy (accountability, inclusion), pink for institutional quality (gov. effectiveness, rule of law), orange for peace breakdown (conflict history, local conflict), gray for baseline.

and test periods; and (ix) excluded extreme outliers and influential countries. We further test the robustness of our estimation strategy by assessing the contribution of political development indicators in predicting flood mortality via simple random forest algorithms (x), as well as their estimated causal effects in two-way fixed effects Poisson regression models (xi). The results of these tests are documented in Supplementary Figs. S7–S39. The overarching findings from these tests are in line with those reported above; contexts marked by recent severe armed conflict have higher predicted human loss from flooding, and the conflict variables are consistently the best predictors. Accountable and effective institutions also add a valuable contribution to predicting flood mortality, although the relative ranking of the democracy and institutional quality indicators is more sensitive to model and sample specifications. Importantly, political indicators maintain their influence even in a more standard econometric approach relying on two-way fixed effect regression models (Supplementary Fig. S39). Further, models without continent random effects show far stronger predicted effects for the political development variables (Supplementary Figs. S16, S17). Since similar institutions and conflict cluster in specific world regions, this is additional evidence that the estimates from the main specification likely are conservative.

### Counterfactual analysis
These results suggest that even if structural political factors have modest average effects on predicted flood mortality, they may contribute substantively to a society's ability to cope with severe flooding without large-scale loss of life. To illustrate potential real-world implications of these findings, we provide a stylized analysis by comparing out-of-sample predictions relying on observed data against predictions obtained using counterfactual data with manipulated political characteristics. For the latter, we use New Zealand as representative of a democratic, capable, and peaceful society and impose its values recorded in 2018 in our dataset on all other countries in the test set (see Supplementary Section 4 for details). The baseline predictors have identical, observed values in both datasets. Effectively, this test serves to quantify potential avoided flood deaths if the world had achieved SDG 16 by 2015, all else equal.

To that end, we first estimate a model containing all six political predictors on all flood events for the training period (2000–14). We then use results from this model to predict mortality outcomes for each recorded flood event in the test set (2015–18), using observed and counterfactual political predictor values, respectively. Figure 5a shows the out-of-sample 90th percentile posterior predictive distributions for the true and counterfactual samples estimated via Bayesian negative binomial regression. In the observed test set, the model mean predicted mortality per flood event is 92.57. In contrast, the mean prediction for the counterfactual sample, where political characteristics for all countries mimic New Zealand, is nearly a half, at 48.32 lives lost. Figure 5b documents results for additional counterfactual tests where we estimate each of the six political predictors in separate models,

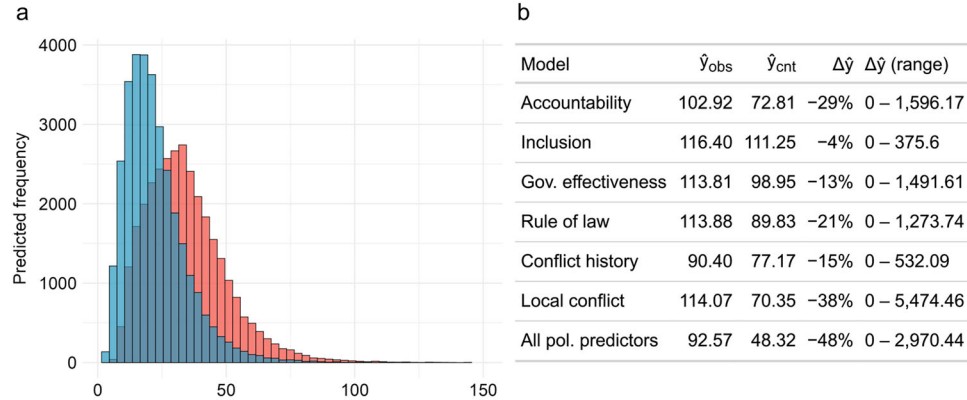

**Fig. 5 | Counterfactual out-of-sample analysis of flood mortality. a** 90[th] percentile posterior predictive distribution of flood mortality across all flood events, 2015–18 ($N = 311$), obtained from models using observed (red) or counterfactual (blue) values on all political predictors. **b** Observed and counterfactual out-of-sample predictions for individual political predictor models and the model containing all predictors. $\hat{y}_{obs}$ represents the average of the mean prediction using observed predictor values; $\hat{y}_{cnt}$ gives corresponding average predictions based on counterfactual predictor values set identical to New Zealand in 2018 in our dataset. **Δŷ** gives the relative difference in ensemble average predictions between observed and counterfactual models. The final column reports the range of difference in the average predicted flood mortality across all 32,000 samples.

| Model | $\hat{y}_{obs}$ | $\hat{y}_{cnt}$ | Δŷ | Δŷ (range) |
| --- | --- | --- | --- | --- |
| Accountability | 102.92 | 72.81 | −29% | 0 − 1,596.17 |
| Inclusion | 116.40 | 111.25 | −4% | 0 − 375.6 |
| Gov. effectiveness | 113.81 | 98.95 | −13% | 0 − 1,491.61 |
| Rule of law | 113.88 | 89.83 | −21% | 0 − 1,273.74 |
| Conflict history | 90.40 | 77.17 | −15% | 0 − 532.09 |
| Local conflict | 114.07 | 70.35 | −38% | 0 − 5,474.46 |
| All pol. predictors | 92.57 | 48.32 | −48% | 0 − 2,970.44 |

similarly to the setup in Figs. 3 and 4. Replacing actual political conditions with New Zealand's values reduces predicted mean flood mortality by 4–48%, depending on the chosen political indicator. Supplementary Fig. S40 shows the results of a similar simulation using the political development of Ghana in 2018 as an alternative and perhaps more achievable benchmark, constituting a peaceful country with high level of political development relative to income level. Although these results come with certain caveats and considerable margin of uncertainty (see Supplementary Section 4 for a discussion), they are indicative of the magnitude of potential gains in disaster risk reduction through political development in non-democratic, weak, and conflict-affected countries.

## Discussion

This research suggests that global flood mortality can be reduced through successful peacebuilding in conflict-affected societies and by improving the institutional quality and accountability of national political institutions. In terms of relative importance, conflict is consistently found to be the most influential political factor. In fact, the predictive contributions of the conflict indicators is very close to those of macroeconomic performance (GDP per capita) and local development (HDI) in terms of marginal conditional effects, and even surpass them for elpd and stacking weights (Figs. 3 and 4), as well as when assessing their added predictive contribution beyond socioeconomic indicators in a joint ensemble (Supplementary Fig. S13). The role of political development in improving disaster preparedness and response clearly extends beyond facilitating socioeconomic growth, underlining the political nature of vulnerability and disaster impacts.

The dam collapse and subsequent flash floods that struck northeast Libya in September 2023, which may have claimed more than 10,000 lives, constitutes a relevant example. Despite exceptional rainfall, experts agree that this was not an unavoidable catastrophe but a result of years of civil war, governance crisis, and negligence of critical infrastructure[83,84]. This stands in stark contrast to, e.g., India and Bangladesh, which have been able to considerably reduce mortality from cyclones and related flooding in recent decades through better disaster preparedness and response, facilitated by more democratic and capable governments[85,86]. Our findings indicate that political development – and in particular the promotion and maintenance of peace – can be an effective way to strengthen societal resilience to natural hazards. In that sense, this study speaks directly to the importance of making progress on SDG 16, and it also offers insights of immediate relevance to the overarching objective of the Sendai Framework for Disaster Risk Reduction[27]. However, fully understanding these pathways requires more granular data than what is currently available, including on the quality of local state and non-state institutions and on bottom-up adaptation measures, such as insurance schemes and community-based disaster risk management[87–89].

While the results documented here carry weight in their own right, the true influence of political development on DRR likely is considerably greater than what we have been able to quantify. For example, high political development not only facilitates greater disaster early warning and response capacity, which reduces human impacts when disaster strikes, but it also incentivizes better protection of citizens and assets from flood exposure in the first place[28]. Both nature-based solutions and structural flood defense systems can reduce the magnitude of inundation or divert flooding away from inhabited land, which may reduce the likelihood that an event would be recorded in our dataset. The baseline control for human exposure hence absorbs and disguises some of the beneficial effect of political development. The problem is likely less prominent for our study than for analyses using strict impact-based data, such as the widely-used EM-DAT disaster database[1], which records hazard events that qualify as "disaster" only (typically entailing loss of life), as the GFD in principle contains information on any significant flood event worldwide regardless of its human and material consequences. DFO is well aligned with observations from river discharge stations and hydrological models[90,91], suggesting a modest selection bias and overall supporting the exogenous nature of our hazard data. Yet, it is clear that, even though accountable and high-quality governance cannot control when and where extreme rainfall and water surge occur, they can help shield major settlements and assets from risk of inundation and destruction due to flooding. We argue that, given such role of political institutions, it is both relevant and necessary to evaluate how institutional factors shape predicted mortality when flooding occurs.

Moreover, information access and quality vary considerably across space and over time. In closed societies and countries with limited media coverage, accurate estimates of disaster exposure and impact are harder to obtain, potentially distorting global statistics such as those provided by DFO. In earlier work, we have identified limitations with flood displacement statistics[76,92]. There is little reason to believe that fatality estimates are immune to these challenges, even if such information generally may be easier to collect and verify than data on elusive outcome categories such as involuntary mobility[93]. For

example, DFO does not provide country-specific casualty estimates for flood events that span across international borders. Our approach, distributing fatalities in accordance with share of population exposure, implies that the analysis likely underestimates the true influence of the contextual predictors on the outcome.

The analysis also revealed challenges related to the relatively short temporal coverage of the flood data (2000–18) and the fact that severe flood impact events (thankfully) are very rare. In order to obtain better out-of-sample prediction, especially for the right tail of the mortality distribution, longer historical records are needed.

These challenges notwithstanding, our research underscores the substantial role of political development in disaster risk reduction, especially in conflict-affected areas or countries with fragile governance. Comprehensively assessing this relationship across the globe, our study verifies and expands on earlier work emphasizing the political nature of disasters[26,42,78,94,95] and constitutes an important counterpoint to reports that quality of governance has little influence on flood fatalities[96]. In contexts of ongoing armed conflict, the central state may not only be unable to address disaster risk deficiencies but even represent an additional hazard to vulnerable communities[94]. While technological upgrading and financial support remain key DRR policies, findings reported here suggest that building peace, justice, and strong institutions in line with SDG 16 is likely to save many lives in the era of climate change. Since higher flood mortality often goes together with widespread human displacement and loss of livelihoods, these results also speak to humanitarian impacts of disasters more generally. Against this backdrop, it is concerning that conflict-affected countries receive comparably little assistance for adapting to climate change[97].

Given complex causal connections and dependencies between different dimensions of political development[28,63], our analysis is unable to establish which specific dimension to prioritize for DRR programming across contexts – other than conflict resolution and peacebuilding to mitigate the impacts of political violence. Future research should aim to disentangle the specific mechanisms through which political factors influence disaster vulnerability and evaluate the effectiveness of common interventions. To this end, there is a clear need for more detailed disaster statistics (e.g., high-resolution spatiotemporal data on within-disaster variation in impact) and better documentation and harmonization of data on disaster risk management plans, mechanisms, and structures. Existing data, such as the FLOPROS flood protection standards database[98] and the Sendai Framework Monitoring System (https://www.desinventar.net/), do not have full global coverage and typically are dependent on self-reporting by governments. Systematic, granular, subnational data on formal and informal institutions, routines, and structures would help to extend and further nuance this analysis, which relies mostly on country-level statistics. Moreover, while there are good reasons to assume that the inverse association between political development and flood mortality documented here is broadly representative across hazard categories, more research is needed on institutional determinants of impacts from other disaster types.

Despite a declining long-term trend in disaster mortality (Fig. 1b, see also ref. [99]), the stagnation in recent years is worrying. Rising global frequency and severity of armed conflict[100], coupled with democratic backsliding in many countries[101], pose clear threats to further progress. Over the coming decades, the growing forces of anthropogenic climate change are projected to result in more severe flooding[102,103]. In this increasingly volatile environmental and geopolitical landscape, it is essential that actors engaged in DRR programming and investments look not only toward technical and financial solutions but also dare address national political barriers to sustainability.

## Methods
### Flood data
The analysis draws exclusively on leading, publicly available data sources. Global data on the prevalence of floods are derived from the Global Flood Database (GFD)[71], which offers high-resolution georeferenced raster images of flooded terrain for the period 2000–18. GFD is based on the Dartmouth Flood Observatory (DFO) archives[72]. DFO codes floods from major media sources and government report, and documents flood events and their impacts on human populations or infrastructure for all years since 1985. GFD matches the flood events recorded by DFO since year 2000 to 250-meter-resolution MODIS satellite data on flooded areas and subjects all flood maps to high-quality control to ensure that only events with clear satellite detection and minimal cloud cover are included. Owing to dense cloud cover and complex topography obstructing remote sensing for many (mostly smaller) events, only a subset of DFO floods recorded since 2000 have been georeferenced[71].

The outcome variable, flood mortality, is based on news reports and information from governmental and non-governmental agencies compiled by DFO. Following the approach in Vestby et al.[76], we split flood events that span across international borders (56.6% of the total) into distinct flood-country events. For these international events, we construct country-specific mortality estimates by dividing the total fatality count among affected countries in accordance with the distribution of population exposure. This approach counts against finding strong effects for the contextual predictors.

Estimates of human exposure to each flood-country event are taken from ref. [76], calculated from spatial overlays between the GFD rasters and high-resolution human settlement data from WorldPop[104]. We drop floods that occurred outside populated areas and events where the exposed population was null, since these events are not at risk of generating loss of life. This returns a valid sample of $N = 2225$ flood-country events, the units of analysis. In sensitivity tests, we estimate the models on alternative samples restricted to fatal flood events ($N = 1070$) and events that exposed at least 1000 people ($N = 1528$), see Supplementary Figs. S20–S23.

All models additionally include a set of flood-specific controls (baseline indicators), imported from DFO: hydrological flood severity, duration of the flood event in days, and the number of previous flood events in the country over the past 10 years. Flood severity is based on the estimated mean return period of each flood event, and includes three severity classes: severity 1 for flood events with an estimated mean return period of 10–20 years; severity 1.5 for events with a mean return period of 21–100 years; severity 2 for extreme events with estimated return period equal to or greater than 100 years[105]. Moreover, we include a dummy for tropical storm-related floods to account for the possibility that casualty estimates for such events are inflated by also including victims of destructive windstorms far outside the flooded areas.

In addition to the flood-specific controls, the baseline includes an index of terrain ruggedness from ref. [77] to capture unobserved topography-related variation in de facto human exposure within flooded areas. The baseline additionally controls for socioeconomic conditions using gross domestic product (GDP) per capita from the World Development Indicators[106] as well as a measure of the weighted average local development in the flood-affected area, calculated by averaging the flooded-area weighted sub-components of the subnational human development index[107] ('Local HDI'). Democracy has a well-documented effect on long-term economic growth[108–110]. The inclusion of socioeconomic variables will hence capture some of the indirect long-term effect of political institutions. We include these variables as they have figured prominently in earlier disaster research, but document results from models without socioeconomic variables as a sensitivity test (Supplementary Figs. S9, S10).

### Predictors
Democratic characteristics of national institutions are represented by two variables. The first is an index of voice and accountability from the Worldwide Governance Indicators (WGI) dataset[35]. Referred to as

'Accountability' in this analysis, the indicator captures the extent to which citizens are allowed to participate in selecting their government, as well as freedom of expression and independence of the media. As a complementary proxy for democracy, we include an indicator measuring degree of equal access to public services among social groups ('Inclusion'), derived from V-Dem[73].

We include two indicators to capture the quality of national political institutions. Bureaucratic competence and effectiveness is represented by a government effectiveness index from WGI[35] that measures, inter alia, the quality of civil services and public goods provision, degree of bureaucratic independence from political pressure, and the government's ability to implement policies ('Gov. effectiveness'). This is supplemented by an index of rule of law, derived from V-Dem[73]. 'Rule of law' reflects the extent to which laws are transparently, independently, predictably, impartially, and equally enforced, and the extent to which government officials comply with the law.

We measure breakdown of peace, the third dimension of political development, by a country-level decay of per capita battle-related deaths (BRDs) from political violence[74], using a half-life parameter of two years to ensure that more recent years weigh more ('Conflict history'). In line with[76], we also include a measure of per capita BRDs within the flooded areas plus a 20 km buffer over the past 12 months prior to flooding outbreak ('Local conflict'). Both indicators are generated from the Uppsala Conflict Data Program's Georeferenced Events Dataset[75].

All continuous political development and baseline predictors, with the exception of the indices, are log-transformed (ln) prior to model estimation to reduce outlier influence. Moreover, all time-varying predictors are lagged by 1 year to mitigate the risk of simultaneity. The specification of lags also increases the efficiency of the estimators in standard regression analyses by attenuating serial correlation in the error terms.

## Statistical analysis

The flood mortality outcome variable exhibits considerable over-dispersion (Supplementary Fig. S1), and the political and socio-economic predictors are correlated (Supplementary Fig. S2) and endogenous, complicating the separation of their individual impacts on the outcome and obstructing the estimation of direct causal effects. Moreover, political factors change only slowly and exhibit limited variation within countries (Supplementary Table S3, Supplementary Fig. S5), such that their within-country variation is not expected to have a meaningful influence on flood mortality within the limits of our 18-year long time-series. Given these constraints, a standard econometric approach relying on country fixed effects would not be suitable for our analysis. Therefore, we rely on predictive performance and evaluation of how indicators of political development inform these predictions in our main empirical strategy, while adding two-way fixed effects only in a limited set of additional models as robustness checks (Supplementary Section 2.12). We assume that a model that provides an accurate representation of the world would yield a stronger predictive performance. Predictive performance is a useful strategy to evaluate the veracity of models, particularly when data present distributional challenges or when predictors are highly correlated[111,112]. Unlike causal inference, which is concerned with internal validity, testing our theoretical argument on unseen data emphasizes external validity[113]. The combination of a substantiated theory based on previous research and its evaluation through both in-sample regression models and out-of-sample predictions can thus overcome the limitations of traditional approaches, while emphasizing generalizability over sample-specific inference.

We employ both in-sample and out-of-sample predictions using Bayesian negative binomial regression models via the BRMS package v2.22.1 in R (v 4.3.2)[114]. This allows us to fit and evaluate models that

provide probabilistic predictions, simulated over 8 × 4000 Markov chain Monte Carlo samples. We relied on default, weakly informative priors to ensure flexibility without overfitting. Convergence diagnostics (Supplementary Fig. S6) indicate reliable posterior estimates across all models. A probabilistic approach is particularly suitable to study the relationship of interest, which is complex and characterized by an over-dispersed outcome, as well as highly correlated predictors whose effects may be difficult to isolate empirically. Probabilistic predictions also allow us to explore how the models perform in the right tail (analogous to estimation of worst-case outcomes): this is particularly important, as evaluation of point predictions can be misguided when the predicted outcome distribution has a heavy tail[115]. Given the complexity of the relationship examined in our study and the data utilized to explore it, we expect that the structural political variables do not necessarily change the median level of fatalities but rather that high political development makes mass mortality from disaster events less likely. Relative to standard approaches that could capture the effect of political development on average flood mortality, a probabilistic approach such as the Bayesian models employed here inherently represents the uncertainty around the predictions, providing a more comprehensive representation of the role of political factors in affecting human costs of flood. By modeling uncertainty in the estimation process, this approach evaluates the model predictive performance across the full predictive distribution, and consistently reflects whether the covariates improve our ability to predict unlikely but severe outcomes. As wider predictive intervals also correspond to higher uncertainty, posterior distributions thus offer a more realistic representation of our confidence in the models' results than the ones provided by traditional models based on point estimates.

We first estimate a linear baseline model, consisting of flood hazard characteristics (population exposure, flood severity, flood duration, historical flood frequency, and tropical-storm related flood), rugged terrain, and socioeconomic conditions (GDP per capita and local HDI). These variables are included in all models documented in Figs. 3–5. Next, we estimate separate models for each of the six indicators for political development, which are added as linear terms to the baseline specification in order to assess their individual contributions to model performance. In Supplementary Information, we also consider the performance of the widely used electoral democracy index from V-Dem[73] as a potential catch-all proxy for political development (Supplementary Fig. S8). All main models are specified with continent-level random intercept and random slope for the number of floods events over the past decade as well as a linear yearly trend, all of which are found to improve models' fit and performance and jointly ensure more reliable estimates. Including continent-level random effects account, at least partly, for unobserved heterogeneity at the regional level that might otherwise lead to omitted variable bias.

In Supplementary Figs. S9, S10, we document results for models that omit controls for the national and local socioeconomic context as these predictors can be considered intermediate drivers that attenuate the estimated influence of political factors on disaster vulnerability[63,64]. Additionally, in Supplementary Table S6 and Supplementary Fig. S39, we present results for a set of two-way fixed effects Poisson regression models, where each political indicator is added to the baseline variables. In line with our main models, the results show that accountable and effective institutions and lack of political violence decrease flood deaths on average, and their effect is statistically significant.

## Assessment of predictive performance

To evaluate variable importance and model performance, we split the total sample of 2225 flood-country events into training and test samples. The training sample contains all flood-country observations during the years 2000–14 ($N = 1914$) whereas the test set covers all observations for 2015–18 ($N = 311$). We alter the training and test periods in sensitivity tests (Supplementary Figs. S24, S25). Both in-sample

and out-of-sample, we estimate the expected log predictive density (elpd) of the posterior distribution using leave-one-out cross-validation and Pareto smoothed importance sampling (PSIS). We choose this evaluation metric since the outcome variable, flood mortality, is highly over-dispersed, rendering common point prediction metrics such as the mean square error (MSE) unsuitable[81,115]. The elpd is proportional to the MSE for normally distributed data with constant variance, and larger values (i.e., values closer to 0) imply better predictive performance.

As a complementary metric for model performance, we use stacking weights. Whereas elpd quantifies accuracy across all observations in the prediction sample, stacking privileges complementarity by assigning weights to the models according to their relative contribution to the optimal combination of models in the ensemble. If one model is better than all others across all observations, it will get the full weight (score 1), regardless of the margin to the next best model. If a model offers the best prediction for 30% of the observations, it will get a score of 0.3. The stacking weights are estimated on a set of K models and sum to 1.

### Reporting summary
Further information on research design is available in the Nature Portfolio Reporting Summary linked to this article.

## Data availability
All raw input data are freely available, and original sources are indicated in the Methods section. Information on floods worldwide, including flood severity, duration, tropical storm-related floods, and count of past flood events, is extracted from the Global Flood Database, which is available from: https://global-flood-database.cloudtostreet.ai/. Gridded data on population exposed to flooding and local conflict in the flooded area were prepared by Jonas Vestby and co-authors for their article "Societal determinants of flood-induced displacement" published in PNAS 121 (3), 2024, and can be downloaded from: https://dataverse.harvard.edu/dataset.xhtml?persistentId=doi:10.7910/DVN/JMAP2M. Dartmouth Flood Observatory (DFO) data on flood mortality are available from: https://floodobservatory.colorado.edu/Archives/. Country-year data on governmental accountability and effectiveness are provided by the Worldwide Governance Indicators (v2023) and are available from: https://www.worldbank.org/en/publication/worldwide-governance-indicators. Country-year indicators of inclusion and rule of law are provided by V-Dem (v13) and can be downloaded from: https://www.v-dem.net/data/the-v-dem-dataset/. Geo-referenced data on armed conflict are available from the Uppsala Conflict Data Program: https://ucdp.uu.se/downloads/. Country-level data on terrain ruggedness are provided by Nathan Nunn and Diego Puga for their article "Ruggedness: The blessing of bad geography in Africa" published in the Review of Economics and Statistics 94(1), 2012, and can be downloaded from: https://diegopuga.org/data/rugged/. Country-year statistics on Gross Domestic Product are available from the World Bank's World Development Indicators at: https://databank.worldbank.org/source/world-development-indicators. Sub-national level data on local HDI are provided by the Global Data Lab and can be downloaded from: https://globaldatalab.org/shdi/. CShapes GIS data of country boundaries are available from https://cran.r-project.org/web/packages/cshapes/index.html. Replication data to reproduce the results reported here, which were created from these sources, have been deposited in Harvard Dataverse with accession code https://doi.org/10.7910/DVN/YMUDX8.

## Code availability
Code to reproduce the analysis[116] is available at https://github.com/prio-data/political_development_flood and archived at https://doi.org/10.5281/zenodo.17314421117.

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

## Acknowledgements

We are grateful to colleagues at PRIO, Uppsala University, University of Konstanz, participants at the 2023 Conflict Research Society annual conference (London, UK), the 2024 Climate Change Adaptation and Mitigation Conflicts workshop (Braunschweig, Germany), the 2024 American Political Science Association convention (Philadelphia, USA), the CC2C Climate Conflict webinar (Florence/Milan, Italy), and members of the Nordic Climate Conflict seminar series for comments on earlier drafts. This research was funded by the European Union through European Research Council (ERC) grant no.

101055133 (www.prio.org/projects/polimpact) awarded to H.B. Views and opinions expressed are however those of the authors only and do not necessarily reflect those of the European Union or the ERC. Neither the European Union nor the granting authority can be held responsible for them.

## Author contributions

Conceptualization: P.V., N.U., H.B.; Data and methodology: P.V., J.V.; Analysis: P.V.; Visualization: P.V., H.B.; Writing: P.V., N.U., H.B.; Coordination: H.B.

## Competing interests

The authors declare no competing interests.
