## [Transparent Peer Review file · Nature Communications]

Political development predicts reduced human cost of flooding

Corresponding Author: Dr Paola Vesco

Version 0:

Reviewer comments:

Reviewer #1

(Remarks to the Author)

This paper analyzes the role of macro-level political development in shaping disaster risk, with a global perspective and by analyzing flood mortality records for flood events recorded between 2000 and 2018. The paper is very interesting, timely in the literature, and well written. I enjoyed reading it. I have some suggestions for improvement aimed at enhancing clarity and consistency, which I write in detail here below. Most of my comments are minor suggestions and can be addressed with amendments to the paper text, but some of them on the methods and how socio-economic characteristics are included in the current analysis might require additional analysis/modelling:

- Title: the title of this paper currently refers to “Political development” as the main factor that can reduce human cost of flooding. The results described in the paper confirm this statement. However, it's striking to see how “Conflict history” and “Local conflict” emerge as the most important variables from the numerical analysis. The authors also mention often in the paper that the presence of (frequent and severe) conflicts is the main threat to further progress in reducing flood vulnerability. This seems to me somehow more specific than “political development”, since political development includes conflicts, decision making processes, bureaucratic capacity, etc. I would accordingly suggest tailoring the title in a way that the role of conflicts is more explicitly mentioned, for better consistency with the main findings of this paper.
- Context: The authors often refer to “(formal) political institutions” in the paper, starting at line 69. However, it is never defined in the paper which institutions can be considered relevant political institutions in the context of this paper and flood vulnerability. Are the authors primarily/only referring to national political institutions (national government)? Are regional and local governmental institutions also included? Better defining what a political institution of interest is in the scope of this paper would facilitate understanding and context to a broad range of readers.
- Context: The authors mention the “three dimensions of political development” in line 76. I assume that these three dimensions are the ones mentioned in line 69 also, i.e., democratic characteristics of formal political institutions, capacity of performance of these institutions, and peaceful modes of conflict resolution. However, in the paragraph starting in line 76, they use different terminology: “accountable and inclusive power-sharing”, “transparent decision-making”, “democratic governance structures”, etc. For clarity and consistency, I would suggest keeping the terminology consistent and explicitly mention to which of the three dimensions of political development the elements mentioned in lines 76-92 belong to.
- Methods: Why did the authors decide to use flood events in the period 2000-2014 for training and those in the period 2015-2018 for testing, instead of shuffling the data? Since there are trends in weather-related disasters and accuracy/completeness of reporting (e.g., those represented in Fig. 1), shuffling the data might avoid biases that may potentially lead to very different data distributions between training and test data.
- Methods: The authors have mentioned several times in the paper that socioeconomic features are relevant and also connected to political features. However, the only socioeconomic variables considered in the sensitivity tests are the GDP per capita and local HDI. Recent work has shown that local socio-economic characteristics are relevant to understand access to flood adaptation measures, proactive behaviors, and potential inequalities. For instance, this is shown in “Veigel, N., Kreibich, H. and Cominola, A., 2023. Interpretable machine learning reveals potential to overcome reactive flood adaptation in the continental US. *Earth's Future*, 11(9), p.e2023EF003571.”. Would it be feasible, at the global scale that the authors are considering, to add more socio-economic features related, e.g., to population density, education level, inequalities (e.g., the Gini coefficient)? I believe that more variables in this sense would allow better comparing top-down effects (political dimension) with bottom-up characteristics (socio-economic characteristics of a population).
- Methods and Discussion: Following on the previous comment, recent papers (including Veigel et al., 2023 mentioned above) also highlight the importance of private precautionary to mitigate the impact of extreme events like floods. The access

to bottom-up private precaution measures (by individuals) is also very much dependent on top-down local conditions (e.g., incentive programs, such as the Community Rating System for flood insurance purchase in the USA). How are the dimensions of (i) private precautionary measures and (ii) incentives/limitations to individuals' access to these measures considered in this study, along with potential changes over time? For sure the political conditions facilitate/limit access to these measures, but as far as I can see the interplay between the political context (top-down) and individuals' flood adaptation actions (bottom-up) are not currently analyzed.

• In the discussion, I really appreciate that the authors mention one specific example for one important flood event – the 2023 flash flood in northeast Libya. This is a very good example to enhance interpretability of the overall aggregated, global, results presented in the paper. I would encourage the authors to add other meaningful examples like this to better support understanding and interpretation of the results they obtained, either in the main paper (if paper length allows) or in the Supplementary Materials.

Reviewer #2

(Remarks to the Author)

This paper uses global datasets of flood mortality supplemented by political indicators and flood characteristics to answer the question how political development, especially conflict, influences the social outcome of flooding. Its goal is quite ambitious. The statistical analyses seem to be compelling. However, I have some questions/concerns about the analyses.

1. The authors briefly gave the rationale of not using standard econometric approach, which is that the flood mortality outcome variable exhibiting over-dispersion and political and socioeconomic predictors are correlated. But the authors did not specifically discuss the suitability/superiority of using in-sample and out-of-sample prediction Bayesian negative binomial regression models. There are so many alternative analytical strategies out there. Why this particular method? If prediction rather than explanation is the main goal for the paper, why not use machine learning methods that can do a good job predicting?

2. Based on the description of the Methods, it seems that there are some temporal discrepancies between these variables. The analysis covers a long period of time from 2000 - 18. It is unclear whether the variables representing democratic characteristics, the quality of national political institutions are constant throughout this period or different. The third dimension of political development, a country-level decay of battle-related deaths from intrastate conflict, however, is taken during the previous calendar year. The authors need to provide more description of the nature of the data and discuss any potential limitation brought by the data.

3. The outcome variable, country-specific mortality, is constructed by dividing the total fatality count among affected countries in accordance with the distribution of population exposure. This procedure of creating this outcome variable is problematic. The operation itself is based on the assumption that the fatality is distributed among affected countries proportional to population exposure. This is as if the whole analysis is not needed because fatality is the direct result of population exposure. Another big issue with this outcome variable is that it does not consider temporal autocorrelation as one country may have experienced multiple flood events during this study period. Time is not considered in the statistical analysis.

Reviewer #3

(Remarks to the Author)

Comments on “Political development reduces human cost of flooding”

This paper empirically examines the role of political development, measured by a variety of indicators related to democracy, institutional quality and performance, and peace, in influencing a country's capacity to cope with natural disasters. Using Bayesian prediction model, the authors find that these factors can significantly reduce fatalities from flooding, and conflict has shown particularly strong predictor power. Their findings highlight the importance of political institutions for mitigating disaster losses and adaptation to climate change. Overall, this research addresses an important question related to the social determinants of disaster losses. The conceptual arguments are plausible and the analysis seems solid. Nonetheless, I have some serious concerns about the novelty of this research paper and its empirical design, which are discussed below.

First of all, the authors on this paper have missed a prominent body of literature examining the effect of economic development and political institutions on disaster losses (e.g., Kahn 2005; Anbarci et al. 2005; Keefer, Neumayer, & Plumper, 2011; Kellenberg & Mobarak, 2008; Rashky, 2008; Toya & Skidmore, 2007; Fankhauser & McDermott, 2014). Most of these papers have used cross-country samples and econometric modeling to explore the social factors influencing disaster losses, including both fatalities and economic damage. The authors may want to pay particular attention to the study conducted by Ferreira, Hamilton, and Vincent (2013) which also used the DFO data to analyze the effect of income and governance on flood-induced fatalities. Their results suggest that better governance has limited effect on reducing fatalities during flood events when controlling for time-invariant unobserved cross-country heterogeneity. The authors should discuss these existing studies in their paper and also clearly articulate the intended contributions of their study. Since this research question is not new, what aspects of this study would contribute to its novelty?

Related to the first point, I am also concerned with the predictive modeling used in this paper. The authors were correct in discussing the endogeneity problems, but their current methodology does not seem to address the endogeneity issue but rather circumvent it. I am not fully convinced that their empirical approach can adequately address the potential omitted variable bias (for example, any unobserved local capacity, culture, and social capital that may correlate with political

development and simultaneously affect flood fatalities) and can be really informative. More discussion is needed to explain and justify their modeling approach, why it has advantage over other approaches used in the literature, and also, for instance, how the Bayesian negative binomial model differs from the standard negative binomial model. Moreover, while the authors noted that this paper focuses on the predictive aspect, they have a relatively narrow focus on comparing the predictive power of different political institutional variables. It would be worth comparing them with other variables (e.g., flooding magnitude, human exposure, and previous flooding events) in terms of predictive power. Presenting these results in the main paper will provide a clearer picture of how much political development matters (relative to other factors). I also suggest the authors include socioeconomic variables, such as GDP per capita, in the baseline model because previous research typically accounts for both incomes and institutions both. It would also be more informative to compare the predictive power of incomes/economic development and that of political institutions.

Regarding the flooding data, the authors may want to provide more details about how a flooding event is defined and included in the GFD. The authors are correct in noting that flood events can be endogenous to societies' underlying vulnerability, and this can be problematic for EM-DAT which included a disaster event based on its losses. However, the paper still does not clearly explain how DFO or GFD records a flooding event or their selection criteria. I do not necessarily believe that flooding damage should be modeled based on an event basis, as its occurrence is also socially determined by factors such as local flood control infrastructure. While earthquakes can be treated as discrete events, flood cannot. Extreme precipitation may trigger flooding in certain areas, but this is not always the case in places with adequate mitigating capacity. I suggest the authors create a balanced panel dataset of the number of flooding events by country-year, and empirically test the correlation between political institutions and flood counts.

Some other minor things:

1. Need to explain how the hydrological flood severity variable is measured/constructed by DFO.
2. What is the percentage of the flooding events that span across international borders in the data set?

References:

Anbarci, N., Escaleras, M., & Register C. A. (2005). Earthquake fatalities: The interaction of nature and political economy. *Journal of Public Economics*, 89, 1907–1933.

Fankhauser, S., & McDermott, T. K. J. (2014). Understanding the adaptation deficit: Why are poor countries more vulnerable to climate events than rich countries? *Global Environmental Change*, 27, 9–18.

Ferreira, S., Hamilton, K., & Vincent, J. R. (2013). Does development reduce fatalities from natural disasters? New evidence for floods. *Environment and Development Economics*, 18(06), 649–679.

Kahn, M. (2005). The death toll from natural disasters: The role of income, geography and institutions. *Review of Economics and Statistics*, 87(2), 271–284.

Keefer, P., Neumayer, E., & Plumper, T. (2011). Earthquake propensity and the politics of mortality prevention. *World Development*, 39(9), 1530–1541.

Kellenberg, D. K., & Mobarak, A. M. (2008). Does rising income increase or decrease damage risk from natural disasters? *Journal of Urban Economics*, 63, 788–802.

Rashky, P. (2008). Institutions and the losses from natural disasters. *Natural Hazards and Earth System Sciences*, 8, 627–634.

Toya, H., & Skidmore, M. (2007). Economic development and the impact of natural disasters. *Economics Letters*, 94, 20–25.

Version 1:

Reviewer comments:

Reviewer #1

(Remarks to the Author)

After checking the reply to the reviewers and the revised manuscript, I believe that the authors have well addressed all my comments. Therefore, I recommend accepting this paper for publication.

(Remarks on code availability)

I have quickly checked the repository and code, but did not run it. Overall, the repository seems tidy and easy to navigate.

Reviewer #3

(Remarks to the Author)

Comments on "Political development predicts reduced human cost of flooding" (revised manuscript)

I would like to commend the authors for their efforts in revising the paper and addressing the reviewers' comments. The revised manuscript shows improvements in clarity and the rigor of the analysis. However, I still have two major concerns regarding their responses to my previous comments.

First, I still believe that the authors should include economic variables such as GDP per capita in their baseline model, rather than just incorporating the economic variable in a robustness check. The authors noted a smaller magnitude of the political development indicators in their additional test including GDP per capita and FDI. This suggests that omitting economic variables leads to an overestimation of the effect of political development variables, as the latter appears to pick up the impact of GDP per capita. Additionally, the authors did not compare the relative influence of economic development and political development variables on flood fatalities. I also am not convinced that including the economic variable would be "bad controls". The bad control problem usually arises when variables that are themselves affected by the treatment (or independent variable of interest), also known as post-treatment variable, are included as controls in a regression model. I am not sure whether the political variables should be considered as clear "treatment" here considering its limited variation over time within a country. Even if that is the case, the authors could consider including GDP per capita for the same year t as the political variables or even using lagged GDP measures from prior years. It is important to note that while GDP and political-economic variables may be highly correlated, they each capture distinct aspects of a country's capacity to manage disaster risks and reduce disaster-induced fatalities. That is why the existing research modeling disaster losses using cross-country samples typically includes both economic and political variables to avoid the omitted variable bias. More importantly, even if both political and economic variables are included in the same model, their model (which includes continent-level random effects) still cannot control for the unobserved cross-country heterogeneity that influence disaster losses. I assume that this is the main reason why their findings differ from Ferreira et al. (2013).

Second, I am not convinced by the authors' reasoning of not modeling the flood frequency or probabilities. Using the DFO data, the paper models flooding fatalities based on observed events, but excludes situations where flooding could have occurred due to extreme weather shocks but was prevented by stronger political institutions. Specifically, Kahn (2005) found that countries with higher incomes have a statistically lower probability of experiencing flooding. This suggests that only modeling those observed/recorded flooding events may cause a sample selection bias. The authors should be more explicit about their rationale for not addressing this particularly problem.

Reference:

Matthew E. Kahn; The Death Toll from Natural Disasters: The Role of Income, Geography, and Institutions. *The Review of Economics and Statistics* 2005; 87 (2): 271–284. doi: <https://doi.org/10.1162/0034653053970339>

(Remarks on code availability)

Reviewer #4

(Remarks to the Author)

The paper investigates the influence of political developments, namely democracy, institutional quality and conflict, on flood mortality using global datasets. As I was specifically asked to comment on the methodological questions raised by reviewer 2 and the response by the authors, I will limit my review to those questions.

Overall, the study is very insightful and the statistical analysis is comprehensive and well-executed including various robustness checks. Reviewer 2 challenged the rationale of the methodological approach using Bayesian regression models over conventional econometric approaches or other predictive approaches (R2.1). While the authors responded with an additional sensitivity test by training another predictive model (random forest), I think the more important part of the question posed by Reviewer 2 is why the authors chose a predictive over an inferential modelling approach.

On page 14/L451f the authors argue that they 'assume that the accurate causal model would exhibit strong predictive performance, implying that any theoretically informed model demonstrating higher predictive accuracy in this setup is likely to be closer to the true causal model and better suited to reflect the true data generation process.' However, I do not think this is a sensible assumption and is in my opinion also not backed by the referenced paper by Cranmer & Desmarais. That paper suggests that predictive models can complement theoretical concepts but not replace them. If the paper is interested in operationalising a theoretical model/concept (as it currently suggests) then a traditional inferential approach should be preferred. Predictive models could form an interesting additional confirmation as suggested by Cranmer & Desmarais but cannot replace an inferential analysis. If the authors' main goal is to successfully predict mortality, this should be made clear in the paper. The choice between an inferential and predictive approach is also not directly linked to a Bayesian vs a frequentist approach and the authors could additionally present regression coefficients of their Bayesian model with the same methodological advantages in regards to data distribution and dealing with small sample sizes. Those would also be more intuitive to interpret than elpds.

Other than that, the paper makes an interesting and relevant contribution.

(Remarks on code availability)

Version 2:

Reviewer comments:

Reviewer #3

(Remarks to the Author)

I am satisfied with the changes the authors have made in this revised manuscript.

(Remarks on code availability)

Reviewer #4

(Remarks to the Author)

The authors made considerable improvements to justify their choice of using a Bayesian predictive modelling approach over a 'conventional' fixed-effects regression model. The main point the authors have now made clear is that their methodological choice is the result of limitations in the available data and not necessarily a superior approach in general. Given that the results are novel, interesting and robust within the clearly outlined limitations, I would recommend the manuscript to be published.

(Remarks on code availability)

Revision memo: ‘Political development predicts reduced human cost of flooding’

Reviewer #1

This paper analyzes the role of macro-level political development in shaping disaster risk, with a global perspective and by analyzing flood mortality records for flood events recorded between 2000 and 2018. The paper is very interesting, timely in the literature, and well written. I enjoyed reading it. I have some suggestions for improvement aimed at enhancing clarity and consistency, which I write in detail here below. Most of my comments are minor suggestions and can be addressed with amendments to the paper text, but some of them on the methods and how socio-economic characteristics are included in the current analysis might require additional analysis/modelling:

R1.1. Title: the title of this paper currently refers to “Political development” as the main factor that can reduce human cost of flooding. The results described in the paper confirm this statement. However, it’s striking to see how “Conflict history” and “Local conflict” emerge as the most important variables from the numerical analysis. The authors also mention often in the paper that the presence of (frequent and severe) conflicts is the main threat to further progress in reducing flood vulnerability. This seems to me somehow more specific than “political development”, since political development includes conflicts, decision making processes, bureaucratic capacity, etc. I would accordingly suggest tailoring the title in a way that the role of conflicts is more explicitly mentioned, for better consistency with the main findings of this paper.

Response: We thank the reviewer for this point. It is true that we find that conflict has a greater effect on predicted flood mortality than the other dimensions of political development, although the latter effects arguably are non-negligible. Moreover, non-democratic and low-quality political institutions are more prevalent worldwide than is armed conflict, implying that the effects of these variables are more relevant and representative for the majority of countries. The revised manuscript (e.g., lines 164-170 and the counterfactual analysis) now highlights more clearly that democratic characteristics and quality of institutions have a meaningful influence on predicted flood mortality at the higher end of the distribution (the upper end of the 80% predictive interval in the conditional effects plot in Fig. 3) despite a more modest median effect. The stylized counterfactual analysis (Fig. 5) provides additional evidence that all tested indicators for political development matter in reducing global flood mortality risk. Given these revisions, we have opted for keeping the original, broader focus, but to address some issues raised by the reviewers concerning endogeneity and to better reflect the nature of the analysis, we have revised the title so it now reads “Political development predicts reduced human cost of flooding”.

R1.2. Context: The authors often refer to “(formal) political institutions” in the paper, starting at line 69. However, it is never defined in the paper which institutions can be considered

relevant political institutions in the context of this paper and flood vulnerability. Are the authors primarily/only referring to national political institutions (national government)? Are regional and local governmental institutions also included? Better defining what a political institution of interest is in the scope of this paper would facilitate understanding and context to a broad range of readers.

Response: Ideally, we would want to account for (theoretically and empirically) institutions and agencies at the local, flood-affected scale as well as national institutions with responsibility and influence over flood risk preparedness and response. Unfortunately, data limitations restrict us to looking into national characteristics only. We have revised the text (lines 63-66 and parts of subsequent paragraphs) to better explain what we mean by formal political institutions. We highlight the limitation of this approach and the need for subnational data on political institutions in the Discussion section of the manuscript (lines 274-277 and 343-345); see also our response to comment R1.6.

R1.3. Context: The authors mention the “three dimensions of political development” in line 76. I assume that these three dimensions are the ones mentioned in line 69 also, i.e., democratic characteristics of formal political institutions, capacity of performance of these institutions, and peaceful modes of conflict resolution. However, in the paragraph starting in line 76, they use different terminology: “accountable and inclusive power-sharing”, “transparent decision-making”, “democratic governance structures”, etc. For clarity and consistency, I would suggest keeping the terminology consistent and explicitly mention to which of the three dimensions of political development the elements mentioned in lines 76-92 belong to.

Response: Thank you for pointing to an area where further clarity was needed. We have updated the relevant text (lines 63-66 and subsequent paragraphs) in line with this suggestion.

R1.4. Methods: Why did the authors decide to use flood events in the period 2000-2014 for training and those in the period 2015-2018 for testing, instead of shuffling the data? Since there are trends in weather-related disasters and accuracy/completeness of reporting (e.g., those represented in Fig. 1), shuffling the data might avoid biases that may potentially lead to very different data distributions between training and test data.

Response: Time-series data such as the ones of interest exhibit inherent temporal dependencies. By reshuffling the data, we may introduce a situation where the model is trained on future data to predict past outcomes, and may thus violate a fundamental principle of time series forecasting by introducing data leakage. Our setup is useful to mimic true future forecasting, where past data is used to train models to predict future

data. This approach is common in predictive modelling and provides a hard test to assess the ability of a model to generalize on unseen data (see, e.g., refs.¹⁻³).

That said, we share the reviewer's concern about different distributions between training and test data. We document results using alternative specifications of the training and test samples in Section 3.8 in the Supplementary Information. Results are comparable to those documented in the main article. Moreover, despite the caveats described above, we have tested reshuffling the data in a random 80-20% split as suggested by the reviewer, and re-run all models as in the main specification. The results of this additional sensitivity test, presented in Section 3.9 of the Supplementary Information, are largely in line with the main findings. The conflict models outperform all models in predicting flood mortality, and accountability has also an important role in predictions both in-sample and out-of-sample (Supplementary Fig. S26). Relative to the main specification, government effectiveness also contributes significantly to the predictive performance in-sample. The conditional effects are also in line with the main specification, although the magnitude of the effect is reduced.

R1.5. Methods: The authors have mentioned several time in the paper that socioeconomic features are relevant and also connected to political features. However, the only socioeconomic variables considered in the sensitivity tests are the GDP per capita and local HDI. Recent work has shown that local socio-economic characteristics are relevant to understand access to flood adaptation measures, proactive behaviors, and potential inequalities. For instance, this is shown in “Veigel, N., Kreibich, H. and Cominola, A., 2023. Interpretable machine learning reveals potential to overcome reactive flood adaptation in the continental US. *Earth's Future*, 11(9), p.e2023EF003571.”. Would it be feasible, at the global scale that the authors are considering, to add more socio-economic features related, e.g., to population density, education level, inequalities (e.g., the Gini coefficient)? I believe that more variables in this sense would allow better comparing top-down effects (political dimension) with bottom-up characteristics (socio-economic characteristics of a population).

Response: We thank the reviewer for this suggestion. We agree that GDP per capita should not be considered a catch-all measure of socioeconomic development, although it is often used to proxy state capacity and macrolevel vulnerability / resilience to climate hazards in the literature. To provide a broader consideration of the socioeconomic context, we also include a local human development index (HDI), which captures levels of education, health, and living standards based largely on population surveys⁴. Indeed, Fig. S9 in the Supplementary Information reveals that the conditional predictive contribution of local HDI is even larger than that of GDP p.c. Moreover, our measure of flood exposure captures important demographic information that shapes mortality levels. In earlier work (e.g., refs.^{5,6}), we have explored the merit of other socioeconomic indicators, including infant mortality rate, subnational inequality indices, and satellite-based nighttime light emissions data. These generally perform similarly to the variables used here. However, we prefer to

supplement GDP p.c. with HDI here as we believe it is more closely aligned with our theoretical understanding of relevant local determinants of vulnerability.

R1.6. Methods and Discussion: Following on the previous comment, recent papers (including Veigel et al., 2023 mentioned above) also highlight the importance of private precautionary to mitigate the impact of extreme events like floods. The access to bottom-up private precaution measures (by individuals) is also very much dependent on top-down local conditions (e.g., incentive programs, such as the Community Rating System for flood insurance purchase in the USA). How are the dimensions of (i) private precautionary measures and (ii) incentives/limitations to individuals' access to these measures considered in this study, along with potential changes over time? For sure the political conditions facilitate/limit access to these measures, but as far as I can see the interplay between the political context (top-down) and individuals' flood adaptation actions (bottom-up) are not currently analyzed.

Response: This is an important observation. Like the reviewer, we believe democracy, high institutional quality, and absence of severe armed conflict facilitate better bottom-up disaster preparedness and response measures, all else equal. Private insurance schemes (where available) are important in this regard, but so are community-based disaster management and response and local governmental agencies^{7,8}. The revised Discussion section (lines 271-277) now briefly acknowledges this point, supported by relevant references (incl. the study mentioned by the reviewer). Although we are not aware of trustworthy time-varying data across our global sample that explicitly measure such bottom-up processes, we would expect these to be at least partly captured by our aggregate political development predictors.

Informed by earlier work (e.g., ref.⁹), we also explore interactions between the political predictors and GDP per capita, since one may expect the roles of democracy and institutional quality in shaping disaster preparedness and response to be contingent on prevailing socioeconomic conditions. As shown in Fig. S11-S12, we indeed find some indication of this: Accountability but also to some extent Government effectiveness become relatively more important predictors in-sample when conditioned on GDP per capita.

R1.7. In the discussion, I really appreciate that the authors mention one specific example for one important flood event – the 2023 flash flood in northeast Libya. This is a very good example to enhance interpretability of the overall aggregated, global, results presented in the paper. I would encourage the authors to add other meaningful examples like this to better support understanding and interpretation of the results they obtained, either in the main paper (if paper length allows) or in the Supplementary Information.

Response: Thank you. We certainly see the merit in offering further real-world validation of our research and now briefly mention Bangladesh and India as examples of successful learning and government-induced investment in inclusive disaster risk reduction (lines 282-285). Moreover, we now also offer a second (and arguably more realistic) counterfactual simulation, assessing the implications of assuming all countries share Ghana's political characteristics (instead of New Zealand's) for global flood mortality, 2015–18. This is mentioned briefly on lines 255-261 and documented in more detail in Section 4 in the Supplementary Information.

Reviewer #2:

This paper uses global datasets of flood mortality supplemented by political indicators and flood characteristics to answer the question how political development, especially conflict, influences the social outcome of flooding. Its goal is quite ambitious. The statistical analyses seem to be compelling. However, I have some questions/concerns about the analyses.

R2.1. The authors briefly gave the rationale of not using standard econometric approach, which is that the flood mortality outcome variable exhibiting over-dispersion and political and socioeconomic predictors are correlated. But the authors did not specifically discuss the suitability/superiority of using in-sample and out-of-sample prediction Bayesian negative binomial regression models. There are so many alternative analytical strategies out there. Why this particular method? If prediction rather than explanation is the main goal for the paper, why not use machine learning methods that can do a good job predicting?

Response: Thank you for allowing us to provide a clearer justification of our research design. This comment also overlaps with R3.2. Our Bayesian approach is motivated by the distributional properties of our data as well as the scarcity of data available for this exercise. A priori, we would not necessarily expect that political development has an important influence on the average level of flood mortality (in the test set, mean flood mortality is 8.3 with median 0), though this is what is evaluated in standard machine learning models. Rather, we expect political factors to matter primarily in reducing mortality from the most extreme events – those displayed at the tail of the distribution. The conditional effect plots in Fig. 3 are consistent with this expectation: although the variables representing democracy and institutional quality have limited *median* effects on predicted flood mortality, the effect is considerably larger at the upper end of the 80% predicted interval (analogous to predicting near-worst case outcome for each event). By using an approach that is fully probabilistic in both estimation and evaluation, we can robustly evaluate the model performance not only at the expectation, but for the full distribution. While doing so is possible also using machine learning approaches (e.g., using pinball loss in random forests), we would be in more uncharted territory in terms of how to properly evaluate these sample distributions (e.g., CRPS is quite sensitive to outliers in heavy-tailed forecast distributions, and we have not seen Pareto-smoothing being used in such applications). Together with the fact that we do not have much data (i.e., a setting

where machine learning excels), we opted for a relatively simple but robust fully distributional approach.

Recent progress in the early warning and prediction literature has highlighted the need to move towards probabilistic forecasts in cases where the outcome is highly skewed and note that simple Negative Binomial models fare well relative to more complex models in these prediction tasks (Hegre et al., *Journal of Peace Research*, *forthcoming*). We have now better motivated our choice of method (lines 456-477) and revised the Results section in places to clarify these nuances in interpreting the findings.

Moreover, to fully address the reviewer's concern, we have run an additional sensitivity test by training a set of random forest models where each political development predictor is added separately to the baseline features (similarly to our preferred specification but without random continent effects as random forests do not allow for that). Note that these models produce predictions as point estimates, i.e., at the expectation. We then compute a measure of importance of each political development indicator based on the random forest models. Specifically, we extract global SHapley Additive exPlanations (SHAP) values as a measure of importance of each political development indicator. SHAP values are based on a coalitional game theory approach and measure the average contribution of each predictor to the overall predictions¹⁰. This approach provides a flexible and consistent measure of global feature importance, similarly to permutation-based feature importance. We present this additional test in Supplementary Fig. S37, and point to this on lines 217-218 in the main manuscript. This test supports the results presented in the manuscript and points to a significant contribution of political development indicators, and conflict variables in particular, in predicting flood mortality.

R2.2. Based on the description of the Methods, it seems that there are some temporal discrepancies between these variables. The analysis covers a long period of time from 2000 - 18. It is unclear whether the variables representing democratic characteristics, the quality of national political institutions are constant throughout this period or different. The third dimension of political development, a country-level decay of battle-related deaths from intrastate conflict, however, is taken during the previous calendar year. The authors need to provide more description of the nature of the data and discuss any potential limitation brought by the data.

Response: We appreciate this comment that points to areas where further clarity is needed. To be clear, all political and socioeconomic predictors are time-varying, typically observed at annual intervals, and imposed a one-year time-lag to minimize the risk of simultaneity (lines 439-443). We have added a new Supplementary Fig. S5, which visualizes global temporal trends for all political development indicators, along with an extensive caption that specifies exactly the indicators used, any

transformation applied before the analysis, as well as their range. Evidently, all indicators exhibit some temporal variation, although the indicators for Accountability and Government effectiveness exhibit relatively less variation than the others indicators in our dataset. The temporal characteristics of the predictors are also briefly mentioned in the Methods section.

R2.3. The outcome variable, country-specific mortality, is constructed by dividing the total fatality count among affected countries in accordance with the distribution of population exposure. This procedure of creating this outcome variable is problematic. The operation itself is based on the assumption that the fatality is distributed among affected countries proportional to population exposure. This is as if the whole analysis is not needed because fatality is the direct result of population exposure. Another big issue with this outcome variable is that it does not consider temporal autocorrelation as one country may have experienced multiple flood events during this study period. Time is not considered in the statistical analysis.

Response: Ideally, we would have data on reported fatalities per flood event in each country, but the statistics provided by DFO do not separate between countries for flood events that span across international borders. In the absence of better information, we distribute reported fatalities across countries in accordance with the distribution of population exposure. This procedure obviously obscures some variation in flood impact but, importantly, it works against finding results for the political variables. Partly for that reason, we believe that our estimates of the role of the contextual predictors are conservative. This is now more clearly communicated in the revised manuscript (lines 309-312 and 378-384).

The reviewer also is right in pointing to temporal dependence in the underlying data. Not all locations are equally flood prone, and impacts from previous flood events may influence the likelihood and severity of future events (e.g., economic losses, material destruction, and overwhelmed response capacity that increase vulnerability to future hazards, but also learning and acceleration of flood protection and early warning measures that reduce vulnerability). Our flood data are not a full panel, as we only have event data on georeferenced flood events. If we had a complete time-series on floods, our modelling strategy would be different. These limitations notwithstanding, we handle such temporal dependencies by (i) accounting for previous flood exposure in the country during the past 10 years, (ii) specifying random slopes that allow for heterogeneity in the effect on past flood exposure on current mortality, and (iii) including a linear time trend. This is described briefly on lines 486-491 in the Methods section, and the performance of these controls is documented in the Supplementary Information, e.g., Fig. S6.

Reviewer #3:

This paper empirically examines the role of political development, measured by a variety of indicators related to democracy, institutional quality and performance, and peace, in influencing a country's capacity to cope with natural disasters. Using Bayesian prediction model, the authors find that these factors can significantly reduce fatalities from flooding, and conflict has shown particularly strong predictor power. Their findings highlight the importance of political institutions for mitigating disaster losses and adaptation to climate change. Overall, this research addresses an important question related to the social determinants of disaster losses. The conceptual arguments are plausible and the analysis seems solid. Nonetheless, I have some serious concerns about the novelty of this research paper and its empirical design, which are discussed below.

R3.1. First of all, the authors on this paper have missed a prominent body of literature examining the effect of economic development and political institutions on disaster losses (e.g., Kahn 2005; Anbarci et al. 2005; Keefer, Neumayer, & Plumper, 2011; Kellenberg & Mobarak, 2008; Rashky, 2008; Toya & Skidmore, 2007; Fankhauser & McDermott, 2014). Most of these papers have used cross-country samples and econometric modeling to explore the social factors influencing disaster losses, including both fatalities and economic damage. The authors may want to pay particular attention to the study conducted by Ferreira, Hamilton, and Vincent (2013) which also used the DFO data to analyze the effect of income and governance on flood-induced fatalities. Their results suggest that better governance has limited effect on reducing fatalities during flood events when controlling for time-invariant unobserved cross-country heterogeneity. The authors should discuss these existing studies in their paper and also clearly articulate the intended contributions of their study. Since this research question is not new, what aspects of this study would contribute to its novelty?

Response: We are grateful to the reviewer for pointing us to additional relevant literature. It is true that our study is not the first to consider political variables in quantitative models of disaster impacts, though we maintain that the influence of specific domestic political characteristics and contexts (e.g., accountable and inclusive institutions, rule of law, violent conflict) is underexplored in the scientific literature and too often ignored in political processes. In combination with important methodological innovation and utilization of high-resolution information on flood characteristics and impacts across a global sample of events, the broad consideration of political determinants of flood impacts (from institutions via capacity to security) imply that our study makes important contributions to scientific understanding of how local and national societal contexts shape flood mortality over and beyond population exposure and hazard severity.

In response to this comment, we now cite several of these additional studies (e.g., lines 321-323) as further motivation of our theoretical argument and to highlight how our study advances the field. Compared to Ferreira et al. (2013) mentioned by the reviewer, our study stands out in several ways that likely explain diverging conclusions. Perhaps most importantly, our Bayesian approach (see also responses to comment R2.1 and 3.2) enables us to move beyond considering average treatment

effects to explore the importance of covariates in predicting the full distribution of flood outcomes. As reported in the manuscript, we find that democracy has a modest effect on the average flood but is more important in curbing severe outcomes. Our study also includes more recent flood events, makes use of satellite imagery data to more accurately detect and account for local characteristics of flood-affected areas, and we use in-sample and out-of-sample prediction to assess variable importance, given inherent challenges with endogeneity, multicollinearity, and likely complex causal pathways that challenge inference from conventional econometric approaches.

R3.2. Related to the first point, I am also concerned with the predictive modeling used in this paper. The authors were correct in discussing the endogeneity problems, but their current methodology does not seem to address the endogeneity issue but rather circumvent it. I am not fully convinced that their empirical approach can adequately address the potential omitted variable bias (for example, any unobserved local capacity, culture, and social capital that may correlate with political development and simultaneously affect flood fatalities) and can be really informative. More discussion is needed to explain and justify their modeling approach, why it has advantage over other approaches used in the literature, and also, for instance, how the Bayesian negative binomial model differs from the standard negative binomial model. Moreover, while the authors noted that this paper focuses on the predictive aspect, they have a relatively narrow focus on comparing the predictive power of different political institutional variables. It would be worth comparing them with other variables (e.g., flooding magnitude, human exposure, and previous flooding events) in terms of predictive power. Present these results in the main paper will provide a clearer picture of how much political development matters (relative to other factors). I also suggest the authors include socioeconomic variables, such as GDP per capita, in the baseline model because previous research typically account for both incomes and institutions both. It would also be more informative to compare the predictive power of incomes/economic development and that of political institutions.

Response: This comment overlaps somewhat with R2.1 so we also refer to our response to that comment. It is correctly observed by the reviewer that our approach cannot entirely solve the endogeneity issue. Yet, Bayesian regressions inherently capture the uncertainty around the predicted values. This probabilistic approach enables us to evaluate how political development contributes to predict flood mortality across the entire distribution (not just at the expected value), as well as to provide a more realistic representation of our confidence in the model's results. We have clarified this point in the manuscript (lines 456-477).

Moreover, by including continent-level random effects (both a random intercept and a random slope for the number of past flood events), we can partially account for unobserved heterogeneity at the regional level that might otherwise lead to omitted variable bias. These random effects capture variation between continents, helping to model the influence of unobserved or unmeasured factors that differ across regions. By incorporating these random effects, the model adjusts for latent geographical

influences that could correlate with omitted variables, thus reducing the potential bias in our estimates. We have added a comment on this point in the Methods section (lines 486-491).

Overall, we believe that the combination of a Bayesian approach and a prediction validation both in- and out-of-sample enable us to assess the informational value of political factors in predicting flood mortality. Their predictive power, alongside the conditional effects, offer the strongest basis for understanding flood mortality drivers given the available data. Given the complexity of the data and the interdependencies characterizing the relationship under study, we believe this is the best suited approach. We have now added more detail on the advantage of our approach in the revised manuscript (e.g., lines 180-184 and lines 456-477).

In line with the reviewer's concern, we have also reviewed the text to avoid direct causal language and we have adjusted the title accordingly. Moreover, we have replaced the previous conditional effects plot with an extended figure (Fig. 3) that now also includes the baseline indicators, enabling a comparison of the conditional effects of political development against the baseline predictors as suggested by the reviewer. Unsurprisingly, the number of people directly exposed to flooding is one of the most powerful predictor of mortality. However, the conditional effects for local and national conflict are as strong as, or stronger than, all baseline predictors.

In the Supplementary Information, we present the results of models where the two indicators of socioeconomic development (GDP and local HDI) are included in the baseline (Supplementary Fig. S9-S10), as well as models that additionally include interaction terms between each political development indicator and GDP per capita (Fig. S11-S12). The results of these tests are in line with those presented in the manuscript: Conflict history and local conflict are the best political predictors of flood mortality both in-sample and out-of-sample (Fig. S9). The magnitude of the conditional effects of the political development indicators are smaller for the models that include socioeconomic predictors, but the pattern is otherwise similar to the results presented in the main article (Fig. S10 vs. Fig. 4).

The key reason for not including socioeconomic development in the main specification is concern about 'bad controls'. As political institutions, government performance, and armed conflict have well-documented effects on economic development¹¹⁻¹³, including socioeconomic predictors in the baseline entails adjusting for variables that partly are a consequence of the treatment, thus blocking part of the causal effect of the treatment on the outcome. As discussed, the overarching conclusions of this study are not dependent on the exclusion of GDP per capita and local HDI from the baseline model, so we prefer to keep the original setup with socioeconomic controls only as a sensitivity test. The text has been revised in places to better explain this point (lines 144-147; 269-271; 493-498).

R3.3. Regarding the flooding data, the authors may want to provide more details about how a flooding event is defined and included in the GFD. The authors are correct in noting that flood events can be endogenous to societies' underlying vulnerability, and this can be problematic for EM-DAT which included a disaster event based on its losses. However, the paper still does not clearly explain how DFO or GFD records a flooding event or their selection criteria. I do not necessarily believe that flooding damage should be modeled based on an event basis, as its occurrence is also socially determined by factors such as local flood control infrastructure. While earthquakes can be treated as discrete events, flood cannot. Extreme precipitation may trigger flooding in certain areas, but this is not always the case in places with adequate mitigating capacity. I suggest the authors create a balanced panel dataset of the number of flooding events by country-year, and empirically test the correlation between political institutions and flood counts.

Response: We thank the reviewer for this point. There are three aspects to this comment: First, the reviewer asks us to provide more detail on DFO and GFD data. The original submission provided only a brief presentation of the data and referred to the original DFO and GFD sources for further details. We agree with the reviewer that more information to readers would be helpful. To this end, we have expanded the description of the flood data in the Methods section (lines 361-376).

Second, the reviewer argues that flood occurrence is partly a function of societal characteristics, and that floods therefore cannot be treated as discrete events. We agree on the first point and indeed make the same observation in the Discussion as one reason why we believe our analysis provides a conservative test of the benefit of political development (lines 292-298). But even if peace, institutional quality, and democracy are relevant in preventing people from being exposed to hazards, could these conditions not also facilitate better disaster risk management if settled areas are exposed to flooding? We expect – and find – that this is true, whereby high levels of political development are associated with reduced risk of major loss of life.

Third, the reviewer suggests that we replace the current research design with a country-year correlation analysis with flood count as the dependent variable. From a theoretical standpoint, we struggle to see how that would reveal information that we are seeking, namely the extent to which political conditions moderate vulnerability (in terms of risk of loss of life) to flooding. Neither political development nor latent flood risk is randomly distributed across the globe so any correlation (or lack thereof) between, e.g., democracy and flood rate would say very little about the ability of inclusive and accountable political institutions to lower human cost of flooding. From a methodological standpoint, we do not have a complete dataset on floods - we have a convenience sample of floods. Hence, creating a panel dataset of the number of flood events by country-year would entail filling the existing dataset with many zeros. This would make the predictive task more complex as the distribution of the outcome

variable would become even more skewed than it is currently. The relative ranking of models, however, is unlikely to change.

R3.4. Some other minor things: 1. Need to explain how the hydrological flood severity variable is measured/constructed by DFO. 2. What is the percentage of the flooding events that span across international borders in the data set?

Response: (1) The hydrological flood severity variable is a discrete measure of flood severity based on the estimated mean return period (recurrence interval, or average interval between two events with magnitude equal to or greater than the level concerned, see ref.¹⁴). We have added a clarification on this point in the Methods section (lines 396-400).

(2) The share of flood events that span across international borders in the dataset is 56.6%. This is now stated in the Methods section (line 380).

References cited in the responses

1. Colaresi, M. & Mahmood, Z. Do the robot: Lessons from machine learning to improve conflict forecasting. *Journal of Peace Research* **54**, 193–214 (2017).
2. Hegre, H., Vesco, P. & Colaresi, M. Lessons from an escalation prediction competition. *International Interactions* **48**, 521–554 (2022).
3. Vesco, P. *et al.* United they stand: Findings from an escalation prediction competition. *International Interactions* **48**, 860–896 (2022).
4. Smits, J. & Permanyer, I. The Subnational Human Development Database. *Scientific Data* **6**, 190038 (2019).
5. Vestby, J., Schutte, S., Tollefsen, A. F. & Buhaug, H. Societal determinants of flood-induced displacement. *Proceedings of the National Academy of Sciences* **121**, e2206188120 (2024).
6. Schutte, S., Vestby, J., Carling, J. & Buhaug, H. Climatic conditions are weak predictors of asylum migration. *Nat Commun* **12**, 2067 (2021).

7. Cretney, R. M. Local responses to disaster: The value of community led post disaster response action in a resilience framework. *Disaster Prevention and Management* **25**, 27–40 (2016).
8. Patterson, O., Weil, F. & Patel, K. The role of community in disaster response: Conceptual models. *Popul Res Policy Rev* **29**, 127–141 (2010).
9. Persson, T. A. & Povitkina, M. “Gimme Shelter”: The Role of Democracy and Institutional Quality in Disaster Preparedness. *Political Research Quarterly* **70**, 833–847 (2017).
10. Molnar, C. *Interpretable Machine Learning: A Guide for Making Black Box Models Explainable*. (Christoph Molnar, Munich, Germany, 2022).
11. Acemoglu, D. & Robinson, J. A. *Why Nations Fail: The Origins of Power, Prosperity, and Poverty*. (Cur, New York, 2012).
12. Knutsen, C. H. Democracy, State Capacity, and Economic Growth. *World Development* **43**, 1–18 (2013).
13. de Groot, O. J., Bozzoli, C., Alamir, A. & Brück, T. The global economic burden of violent conflict. *Journal of Peace Research* **59**, 259–276 (2022).
14. Kundzewicz, Z. W., Pińskwar, I. & Brakenridge, G. R. Large floods in Europe, 1985–2009. *Hydrological Sciences Journal* **58**, 1–7 (2013).

**Revision memo,
NCOMMS-24-14894B “Political development predicts reduced human cost of flooding”**

Dear Reviewers,

We are grateful for the opportunity to further refine our study, and we appreciate the constructive feedback from the latest round of peer review. This document recites the comments conveyed in these reviews, accompanied by our point-by-point description of how they have been addressed. Whenever relevant, we point to the corresponding lines in the manuscript (version with track changes).

We hope you find the new version substantively improved.

Sincerely,

The authors

Comments by Reviewer 1

R1.1. After checking the reply to the reviewers and the revised manuscript, I believe that the authors have well addressed all my comments. Therefore, I recommend accepting this paper for publication.

Response: We appreciate the positive overall assessment of our study.

R1.2. I have quickly checked the repository and code, but did not run it. Overall, the repository seems tidy and easy to navigate.

Response: We are reassured that the Reviewer did not identify any flaws in the replication material.

Comments by Reviewer 3

R3.1. I would like to commend the authors for their efforts in revising the paper and addressing the reviewers' comments. The revised manuscript shows improvements in clarity and the rigor of the analysis.

Response: We appreciate the positive overall assessment of our study.

R3.2. However, I still have two major concerns regarding their responses to my previous comments. First, I still believe that the authors should include economic variables such as GDP per capita in their baseline model, rather than just incorporating the economic variable in a robustness check. The authors noted a smaller magnitude of the political development indicators

in their additional test including GDP per capita and FDI. This suggests that omitting economic variables leads to an overestimation of the effect of political development variables, as the latter appears to pick up the impact of GDP per capita. Additionally, the authors did not compare the relative influence of economic development and political development variables on flood fatalities. I also am not convinced that including the economic variable would be “bad controls”. The bad control problem usually arises when variables that are themselves affected by the treatment (or independent variable of interest), also known as post-treatment variable, are included as controls in a regression model. I am not sure whether the political variables should be considered as clear “treatment” here considering its limited variation over time within a country. Even if that is the case, the authors could consider including GDP per capita for the same year t as the political variables or even using lagged GDP measures from prior years. It is important to note that while GDP and political-economic variables may be highly correlated, they each capture distinct aspects of a country's capacity to manage disaster risks and reduce disaster-induced fatalities. That is why the existing research modeling disaster losses using cross-country samples typically includes both economic and political variables to avoid the omitted variable bias. More importantly, even if both political and economic variables are included in the same model, their model (which includes continent-level random effects) still cannot control for the unobserved cross-country heterogeneity that influence disaster losses. I assume that this is the main reason why their findings differ from Ferreira et al. (2013).

Response: Thank you for this constructive feedback. There are several related issues here that we comment on in sequence:

First, we agree with the reviewer that GDP per capita and the political variables capture at least partly distinct features, and we therefore revised the presentation of the analysis such that GDP per capita, as well as local development, now feature in the main specification. This also allows comparing our results more directly to those reported in the literature. We thank the reviewer for encouraging us to rethink this decision.

That said, including economic and political factors in the same model is not without challenges, since there is compelling evidence that the qualities of political institutions have long-term influence on economic stability and growth (e.g., refs 61-64 in manuscript; see also Acemoglu et al., 2019; Colagrossi, Rossignoli & Maggioni, 2020; Knutsen, 2021). Democracy and economic development are not only correlated, they are co-producing each other. Controlling for GDP per capita and local HDI therefore should be considered a hard test for the political variables, whose influence on the outcome is better assessed through metrics of model behavior (e.g., predictive performance) than individual parameter coefficients. We have better clarified these points in the revised manuscript (lines 131-137; 141-145; 187-191; 219-221; 369-375; 546-550).

The previous version of the manuscript included a comparison of the relative importance of the political and economic predictors (lines 269-271 of the previous version). We would like to point to lines 369-375 of the resubmission, which state that the predictive contributions of the conflict indicators is very close to or even surpass those of macroeconomic performance (GDP per capita) and local development (HDI). The revised manuscript offers further details on this. For example, we have run an additional

sensitivity test to assess the added value of conflict beyond socioeconomic indicators (Supplementary Fig. S13). Specifically, we computed the elpd and stacking weights for an ensemble including only two models: a baseline model (with socioeconomic features) and a conflict model where both conflict history and local conflict indicators are added to the baseline features (including socioeconomic variables). This comparison assesses the added value of the conflict indicators in predicting flood mortality beyond socioeconomic variables. The results reveal that the model including conflict predictors has by far the highest contribution to predictive performance both in-sample and out-of-sample, suggesting that conflict indicators contribute higher predictive accuracy than socioeconomic factors.

There are several differences between Ferreira et al. 2013 and our study (e.g., different variable operationalization, data sources, and time periods; quality of georeferencing). Most importantly, that study estimates the effect of short-term changes in democracy, concluding that governance does not affect flood fatalities. We believe the beneficial effects of accountability, inclusion, prevalence of peace, etc. on reducing disaster risk accumulate over time and manifest primarily in the longer term. Given the limited temporal coverage of high-resolution flood data (and thus limited variation in observed political characteristics for many countries), we rejected a conventional causal inference design in favor of testing causal theoretical expectations through assessing out-of-sample predictive performance. This is in line with a prominent literature pointing to the pitfalls of the traditional standard significance testing and strongly recommending a more comprehensive approach which rely on the models' predictive capability and performance on unseen data e.g. (Ward, Greenhill & Bakke, 2010; Lo et al., 2015; Colaresi & Mahmood, 2017; McShane et al., 2019) See also our response to comment R4.2.

R3.3. Second, I am not convinced by the authors' reasoning of not modeling the flood frequency or probabilities. Using the DFO data, the paper models flooding fatalities based on observed events, but excludes situations where flooding could have occurred due to extreme weather shocks but was prevented by stronger political institutions. Specifically, Kahn (2005) found that countries with higher incomes have a statistically lower probability of experiencing flooding. This suggests that only modeling those observed/recorded flooding events may cause a sample selection bias. The authors should be more explicit about their rationale for not addressing this particularly problem.

Response: We agree that flooding can be sensitive to societal characteristics – indeed, this is a key motivation for our study. However, flood risk management is typically less about completely avoiding the overflow of water during extreme rainfall and surge events (which in many cases is unavoidable) than about controlling where flooding occurs and where it does not, protecting critical assets and settlements from harm and thereby reducing or avoiding human losses. This is where we believe good institutions and peace would be particularly relevant and this is why we seek to assess the extent to which these conditions affect predicted mortality when flooding occurs.

To this end, we rely on the Global Flood Database. GFD provides remotely sensed imagery of major inundation events recorded by DFO since year 2000. In contrast to the widely used EM-DAT database, which only includes events that become “disasters” and therefore suffers from the selection effect mentioned by the reviewer, DFO seeks to cover major flood events regardless of their human consequences. Tellingly, thirty percent of DFO floods since year 2000 did not have reported human casualties and the majority caused fewer than five deaths (see also Supplementary Fig. S1). Comparisons of DFO floods with river discharge station data and hydrological models (e.g., Dottori et al., 2016; Najibi & Devineni, 2018) suggest high levels of correspondence, lending credibility to the exogenous nature of our hazard data. Any selection bias in the DFO data therefore is likely to be modest (lines 421-427 in the revised manuscript).

Parenthetically, we also note that the study mentioned by the reviewer (Kahn 2005), using EM-DAT, finds that disasters in higher-income countries tend to be less deadly but not less frequent – a finding that compares well with our results.

We have now expanded our discussion on these caveats and elaborated on how our study mitigates and addresses them in the revised manuscript (lines 158-160; 387-407; 409-427).

Comments by Reviewer 4

R4.1. The paper investigates the influence of political developments, namely democracy, institutional quality and conflict, on flood mortality using global datasets. As I was specifically asked to comment on the methodological questions raised by reviewer 2 and the response by the authors, I will limit my review to those questions. Overall, the study is very insightful and the statistical analysis is comprehensive and well-executed including various robustness checks.

Response: We appreciate the positive overall assessment of our study.

R4.2. Reviewer 2 challenged the rationale of the methodological approach using Bayesian regression models over conventional econometric approaches or other predictive approaches (R2.1). While the authors responded with an additional sensitivity test by training another predictive model (random forest), I think the more important part of the question posed by Reviewer 2 is why the authors chose a predictive over an inferential modelling approach. On page 14/L451f the authors argue that they 'assume that the accurate causal model would exhibit strong predictive performance, implying that any theoretically informed model demonstrating higher predictive accuracy in this setup is likely to be closer to the true causal model and better suited to reflect the true data generation process.' However, I do not think this is a sensible assumption and is in my opinion also not backed by the referenced paper by Cranmer & Desmarais. That paper suggests that predictive models can complement theoretical concepts but not replace them. If the paper is interested in operationalising a theoretical model/concept (as it currently suggests) then a traditional inferential approach should be preferred. Predictive models could form an interesting additional confirmation as suggested by Cranmer & Desmarais but

cannot replace an inferential analysis. If the authors' main goal is to successfully predict mortality, this should be made clear in the paper. The choice between an inferential and predictive approach is also not directly linked to a Bayesian vs a frequentist approach and the authors could additionally present regression coefficients of their Bayesian model with the same methodological advantages in regards to data distribution and dealing with small sample sizes. Those would also be more intuitive to interpret than elpds. Other than that, the paper makes an interesting and relevant contribution.

Response: Thank you for this constructive feedback. We realize that the rationale for adopting a predictive design over conventional causal inference was inadequately presented in the manuscript. In an ideal world, we would rely on inferential analysis to assess the causal contribution of political development to flood mortality. However, as mentioned in our reply to comment R3.2, the available high-resolution flood records (2000-2018) are not sufficiently comprehensive to enable assessing causal (within-unit) effects of structural political variables that tend to change only slowly. As shown in Table M1 (also added as Supplementary Table S3), within-country variation in observed values for the political variables is only a fraction of between-country variation. Specifying country fixed effects would absorb much of the theoretically relevant influence of political development on flood mortality. We elaborate on this point in the revised version of the manuscript (lines 141-145; 603-625).

Besides, nearly half of the countries listed in the GFD database have three or fewer major flood events in the period. Creating a panel dataset to include observations without recorded flooding (ref. comment R3.3) would not solve the short time-series coverage of the flood data, but at the same time would prevent us from making use of flood-specific information (some countries have several flood events in distinct locations in a year, whose local characteristics cannot meaningfully be aggregated to a country-year structure; See also lines 158-160 in revised manuscript).

Instead, we make a case for testing observable implications of a causal theory under the assumption that a model that provides an accurate representation of the world yield strong predictive performance. Predictive performance thus is a way to evaluate the veracity of models, particularly when more robust inferential designs are impossible. Unlike causal inference, which is concerned with internal validity, emphasis here is on external validity by testing expectations on new data (cf. Findley, Kikuta & Denly, 2021). It is the combination of a substantiated theory based on previous research and evaluation through prediction that gives this method merit, not the predictive performance in itself. We have revised the manuscript to clarify the reasoning underpinning our research design and argue more clearly for a predictive design rather than a standard econometric approach (lines 70-73; 141-145; 243-251; 305-306; 369-375; 546-550; 603-625; 678-682).

Table M1. Observed variation in scores for political predictors

	Accountability	Inclusion	Gov. effect.	Rule of law	Confl. history	Local conflict
Within-unit SD	0.13	0.02	0.13	0.03	12.67	116.76
Between-unit SD	0.94	0.26	0.95	0.30	44.97	407.67
Ratio	0.13	0.06	0.14	0.10	0.28	0.29

However, in the interest of complete transparency, we now document results from a set of two-way fixed effects Poisson regression models, where each political indicator is added to the baseline features (see new Supplementary Table S6). The results of these models are in line with the main results documented in the manuscript: conflict history and local conflict has a strong positive effect on flood mortality while accountable and effective institutions are expected to mitigate flood deaths, *ceteris paribus*. These effects are statistically significant ($p < 0.01$). The effects of the other political predictors and of the socioeconomic indicators are in line with expectations, but they are not statistically significant. We briefly discuss these results in the main manuscript (lines 293-294; 305-306; 676-682).

In addition, we have revised the text of the manuscript in places to avoid inappropriate causal language and to make clearer the reasoning behind our predictive approach. These edits are visible in track-changes.

Other implemented changes

In addition to the changes implemented to address reviewer concerns, we have made a few other, smaller revisions. As we included the two additional socioeconomic predictors among the baseline features, we re-trained all models with increased iterations and a higher alpha parameter to ensure better convergence. We have updated the methodology, as well as the presentation of the models' fit accordingly in both the manuscript and the Supplementary Material. We also discovered an error in the original computation of the Local HDI index (due to spatial weighting), which we now fixed. Expectedly, as the indicator is now more precisely measured, the conditional effect for local HDI is slightly higher in the updated results relative to the previous version. We expect the effect of local HDI to at least partly absorb the effect of conflict (especially locally) on flood mortality, due to the influence of violence on socioeconomic development. This expectation seems to be confirmed as the effect of conflict increases when socioeconomic variables are omitted from the baseline (Supplementary Fig. S9).

Lastly, we have slightly edited the text of the Manuscript and Supplementary Material to ensure more clarity where needed.

References

- Acemoglu, Daron, Suresh Naidu, Pascual Restrepo & James A Robinson (2019) Democracy Does Cause Growth. *Journal of Political Economy* 127(1): 47–100.
- Colagrossi, Marco, Domenico Rossignoli & Mario A Maggioni (2020) Does democracy cause growth? A meta-analysis (of 2000 regressions). *European Journal of Political Economy* 61(January): 101824.
- Colaresi, Michael & Zuhair Mahmood (2017) Do the robot: Lessons from machine learning to improve conflict forecasting. *Journal of Peace Research* 54(2): 193–214.
- Dottori, Francesco, Peter Salamon, Alessandra Bianchi, Lorenzo Alfieri, Feyera Aga Hirpa & Luc Feyen (2016) Development and evaluation of a framework for global flood hazard mapping. *Advances in Water Resources* 94(August): 87–102.
- Findley, Michael G, Kyosuke Kikuta & Michael Denly (2021) External Validity. *Annual Review of Political Science* 24(1): 365–393.
- Knutsen, Carl Henrik (2021) A business case for democracy: regime type, growth, and growth volatility. *Democratization* 28(8): 1505–1524.
- Lo, Adeline, Herman Chernoff, Tian Zheng & Shaw-Hwa Lo (2015) Why significant variables aren't automatically good predictors. *Proceedings of the National Academy of Sciences* 112(45): 13892–13897.
- McShane, Blakeley B, David Gal, Andrew Gelman, Christian Robert & Jennifer L Tackett (2019) Abandon Statistical Significance. *The American Statistician* (March) (<https://www.tandfonline.com/doi/abs/10.1080/00031305.2018.1527253>).
- Najibi, Nasser & Naresh Devineni (2018) Recent trends in the frequency and duration of global floods. *Earth System Dynamics* 9(2): 757–783.
- Ward, Michael D, Brian D Greenhill & Kristin M Bakke (2010) The perils of policy by p-value: Predicting civil conflicts. *Journal of Peace Research* 47(4): 363–375.